# Identification of IOMA-class neutralizing antibodies targeting the CD4-binding site on the HIV-1 envelope glycoprotein

Jelle van Schooten [1], Elinaz Farokhi [2], Anna Schorcht[1], Tom L. G. M. van den Kerkhof[1], Hongmei Gao[3], Patricia van der Woude[1], Judith A. Burger[1], Tim G. Rijkhold Meesters [1], Tom Bijl[1], Riham Ghalaiyini[1], Hannah L. Turner[2], Jessica Dorning[4], Barbera D. C. van Schaik[5], Antoine H. C. van Kampen[5], Celia C. Labranche [3], Robyn L. Stanfield[2], Devin Sok[6,7,8,9], David C. Montefiori [3], Dennis R. Burton [6,7,9,10], Michael S. Seaman[4], Gabriel Ozorowski [2,7,8], Ian A. Wilson [2,7,8,11], Rogier W. Sanders [1,12], Andrew B. Ward [2,7,8] & Marit J. van Gils [1] ✉

A major goal of current HIV-1 vaccine design efforts is to induce broadly neutralizing antibodies (bNAbs). The VH1-2-derived bNAb IOMA directed to the CD4-binding site of the HIV-1 envelope glycoprotein is of interest because, unlike the better-known VH1-2-derived VRC01-class bNAbs, it does not require a rare short light chain complementarity-determining region 3 (CDRL3). Here, we describe three IOMA-class NAbs, ACS101-103, with up to 37% breadth, that share many characteristics with IOMA, including an average-length CDRL3. Cryo-electron microscopy revealed that ACS101 shares interactions with those observed with other VH1-2 and VH1-46-class bNAbs, but exhibits a unique binding mode to residues in loop D. Analysis of longitudinal sequences from the patient suggests that a transmitter/founder-virus lacking the N276 glycan might have initiated the development of these NAbs. Together these data strengthen the rationale for germline-targeting vaccination strategies to induce IOMA-class bNAbs and provide a wealth of sequence and structural information to support such strategies.

[1]Department of Medical Microbiology, Amsterdam Infection & Immunity Institute, Amsterdam UMC, Location AMC, University of Amsterdam, Amsterdam 1105AZ, The Netherlands. [2]Department of Integrative Structural and Computational Biology, The Scripps Research Institute, La Jolla, CA 92037, USA. [3]Department of Surgery, Duke University Medical Center, Durham, NC 27710, USA. [4]Center for Virology and Vaccine Research, Beth Israel Deaconess Medical Center, Harvard Medical School, Boston, MA 02215, USA. [5]Bioinformatics Laboratory, Department of Epidemiology and Data Science, Amsterdam Infection & Immunity Institute, Amsterdam UMC, University of Amsterdam, Amsterdam 1105AZ, The Netherlands. [6]Department of Immunology and Microbiology, The Scripps Research Institute, La Jolla, CA 92037, USA. [7]International AIDS Vaccine Initiative Neutralizing Antibody Center, The Scripps Research Institute, La Jolla, CA 92037, USA. [8]Consortium for HIV/AIDS Vaccine Development, The Scripps Research Institute, La Jolla, CA 92037, USA. [9]International AIDS Vaccine Initiative, New York, NY 10004, USA. [10]Ragon Institute of MGH, MIT and Harvard, Cambridge, MA 02139, USA. [11]The Skaggs Institute for Chemical Biology, The Scripps Research Institute, La Jolla, CA 92037, USA. [12]Department of Microbiology and Immunology, Weill Medical College of Cornell University, New York, NY 10065, USA. ✉e-mail: M.J.vangils@amsterdamumc.nl

The human immunodeficiency virus 1 (HIV-1) remains a major public health issue with 37.7 million people living with HIV-1 and 1.5 million new infections in 2020 worldwide[1]. A vaccine is highly desirable to stop the HIV-1/AIDS pandemic and will likely need to induce broadly neutralizing antibodies (bNAbs) that can potently neutralize virus strains from multiple HIV-1 subtypes. bNAbs have been frequently isolated from HIV-1 infected individuals and can target multiple vulnerable epitopes on the HIV-1 envelope glyco-protein complex (Env)[2,3]. Moreover, bNAbs are able to prevent non-human primates from experimental simian-human immuno-deficiency virus (SHIV) infection[4–6], indicating that vaccine-induced bNAbs could also protect humans against HIV-1 infection. Understanding why these bNAbs develop only in some individuals, and more specifically which Env variants have induced these responses, can facilitate the design of HIV-1 vaccines to elicit similar bNAbs.

The CD4-binding site (CD4bs) on the gp120 subunit is an attractive target for HIV-1 immunogen design. The CD4bs is conserved across viral strains as virtually all HIV-1 strains depend on CD4 binding for entry into target cells. Moreover, a substantial number of bNAbs to the CD4bs have been isolated from various HIV-1 infected individuals[2,3], including broad and potent ones such as N6, that achieve near-pan neutralization breadth[7]. Some CD4bs-directed bNAbs mimic the interaction between CD4 and Env by using either a VH1-2 or VH1-46 encoded heavy chain (HC)[8]. The most broad and potent CD4bs bNAbs belong to the VH1-2-derived VRC01-class and are defined by their short five-residue light chain (LC) complementarity-determining region 3 (CDRL3) that interacts with the V5 loop of gp120[9,10]. In addition, VRC01-class bNAbs often require deletions in their CDRL1 and/or an elongated CDRH3 to accommodate the glycan attached to N276 to achieve neutralization breadth and potency[9–12]. The N276 glycan restricts access to the CD4bs and proves a major roadblock for the development of bNAbs to this site.

Various vaccine-design efforts aim to elicit VRC01-class bNAbs by engaging their inferred germline precursor B cells followed by sequential immunization regimens to drive key mutations essential for broadly neutralizing activity[13–17]. However, the rarity of the short CDRL3 in the naïve human B cell repertoire and high degree of somatic hypermutation (SHM) present hurdles for the induction of VRC01-class bNAbs by vaccination. The discovery of the VH1-2*02-, and VL2-23*02-derived IOMA bNAb with an average-length eight-residue CDRL3 and less SHM therefore attracted considerable interest[18]. IOMA-class bNAbs might be easier to elicit since B cells with average-length CDRL3 are present at higher frequency in the naive human B cell repertoire compared to those with a short CDRL3[10]. Secondly, IOMA requires less SHM compared to VRC01-class bNAbs. Thirdly, the IOMA CDRL1 contains amino acid substitutions but does not require deletions as found in a subset of VRC01-class bNAbs[18]. IOMA is however significantly less broad and potent compared to many VRC01-class bNAbs and to date only one class member, i.e., IOMA itself, has been described, limiting the data available for vaccine design. It is therefore unclear to what extent IOMA-class bNAbs have developed in other HIV-1 infected individuals. The identification of additional IOMA-class bNAbs provides valuable information on the genetic and structural properties of this class, as well as the breadth and potency of IOMA-class bNAbs, and hence the potential for this class as a HIV-1 vaccine target. Here, we describe three IOMA-class NAbs, ACS101-103 derived from the VH1-2*02 and VL2-23*02 germlines, that share many prop-erties with IOMA, including an average-length eight-residue CDRL3 and show up to 37% neutralization breadth. These findings indicate that IOMA-class NAbs developed in multiple HIV-1 infected individuals and provide important genetic and structural information for the design of CD4bs-targeting immunogens aimed at inducing IOMA-class bNAbs.

## Results

### Isolation of Env-specific B cells from an HIV-1 infected elite neutralizer

We sorted Env-specific B cells from peripheral blood mononuclear cells (PBMCs) at months 24 and 36 post seroconversion (post SC) from an HIV-1 infected individual, H18877, who was infected with a clade B HIV-1 variant and enrolled in the Amsterdam Cohort Studies on HIV-1/AIDS. Some cross-neutralizing activity was detected in H18877's serum as early as 3 months post SC and within the first year after infection the serum of this individual potently neutralized various viruses from different clades, showing that bNAb responses do not necessarily require years to develop (Fig. 1a)[19]. Individual H18877 was categorized as an HIV-1 infected elite neutralizer as its serum at month 35 post SC neutralized a panel of virus representing the global HIV-1 diversity with a geometric mean of 782 (Fig. 1a)[19,20]. Longitudinal env genes from this patient were sequenced and the consensus sequence of env genes from five viral clones isolated 2 months post SC was used to generate the AMC009 SOSIP trimer[21]. We also generated the autologous virus of one of the five viral clones isolated at 2 months post SC.

PBMC samples of patient H18877 at 24 and 36 months post SC were available for B-cell sorting. B cells from month 24 were sorted with fluorescently labeled clade B AMC009 SOSIP, clade A BG505 SOSIP and clade A 94UG103 gp120 proteins (Fig. S1a). These Env pro-teins were chosen because contemporaneous sera neutralized the parental AMC009, BG505 and 94UG103 viruses potently[20]. Further-more, BG505 SOSIP has been successfully used to isolated bNAbs in previous studies[22–29]. Single B cells that were positive for at least one of the three probes were sorted. B cells from month 36 post SC were sorted with BG505 SOSIP, 94UG103 gp120 and clade MGRM C026 gp120 (Fig. S1b), because contemporaneous sera neutralized the BG505, 94UG103, and MGRM C026 viruses[19,20]. From the month 36 samples, we sorted single Env-specific B cells that could recognize at least one of the three Env proteins. In total, we sorted 370 memory B cells from months 24 and 36 post SC.

### Identification of ACS101-103, three IOMA-class antibodies tar-geting the CD4-binding site

We obtained 222 HC B cell receptor (BCR) sequences from the sorted Env-specific B cells. The sorted B cells used a wide variety of germline genes. Three BCRs, termed ACS101-103, utilized the VH1-2*02 gene (Fig. 1b), a VH1-2 allele that is commonly used by bNAbs targeting the CD4bs, including VRC01 and IOMA[18,30] (Fig. S2). ACS101 originated from PBMCs taken at month 24 post SC, whereas ACS102 and ACS103 were isolated from PBMCs at month 36. We retrieved the corre-sponding LC sequences of ACS101-103 and found that all three BCRs utilized the VL2-23*02 gene segment, i.e. the same one as used by IOMA[18] (Fig. 1c, Fig. S2). Furthermore, all three LCs include an average-length CDRL3 of eight amino acids (Fig. S2). Thus, the genetic make-up of ACS101-103 is very similar as that of IOMA.

ACS101 and ACS103 have an HC complementarity-determining region 3 (CDRH3) of 19 residues, whereas CDRH3 of ACS102 is 16 residues long. ACS103 has a two amino acid insertion in its HC fra-mework region 3 (FWR3), at a similar position as VRC01-class bNAbs 3BNC60 and VRC03[30,31] (Fig. S2). Interestingly, ACS101 and ACS102 contain a two-residue deletion in their CDRL1, similar to VRC01-class bNAbs but not IOMA[18] (Fig. S2). Thus, while ACS101-103 belong to the IOMA class, they also share genetic characteristics with individual VRC01-class members. All three BCRs contain a substantial number of SHMs in their HC (27, 35 and 24 amino acid substitutions for ACS101, ACS102, ACS103, respectively) and LC (18, 16, and 20 substitutions for ACS101, ACS102 and ACS103, respectively) (Fig. 1c, Fig. S2), which is slightly more compared to IOMA (22 and 15 changes in HC and LC, respectively), but less than in VRC01 (41 and 25 changes in HC and LC, respectively) (Fig. S2). All three BCRs share many specific SHM in their HC with IOMA and VRC01, as well as with IOMA's LC but not with the LC

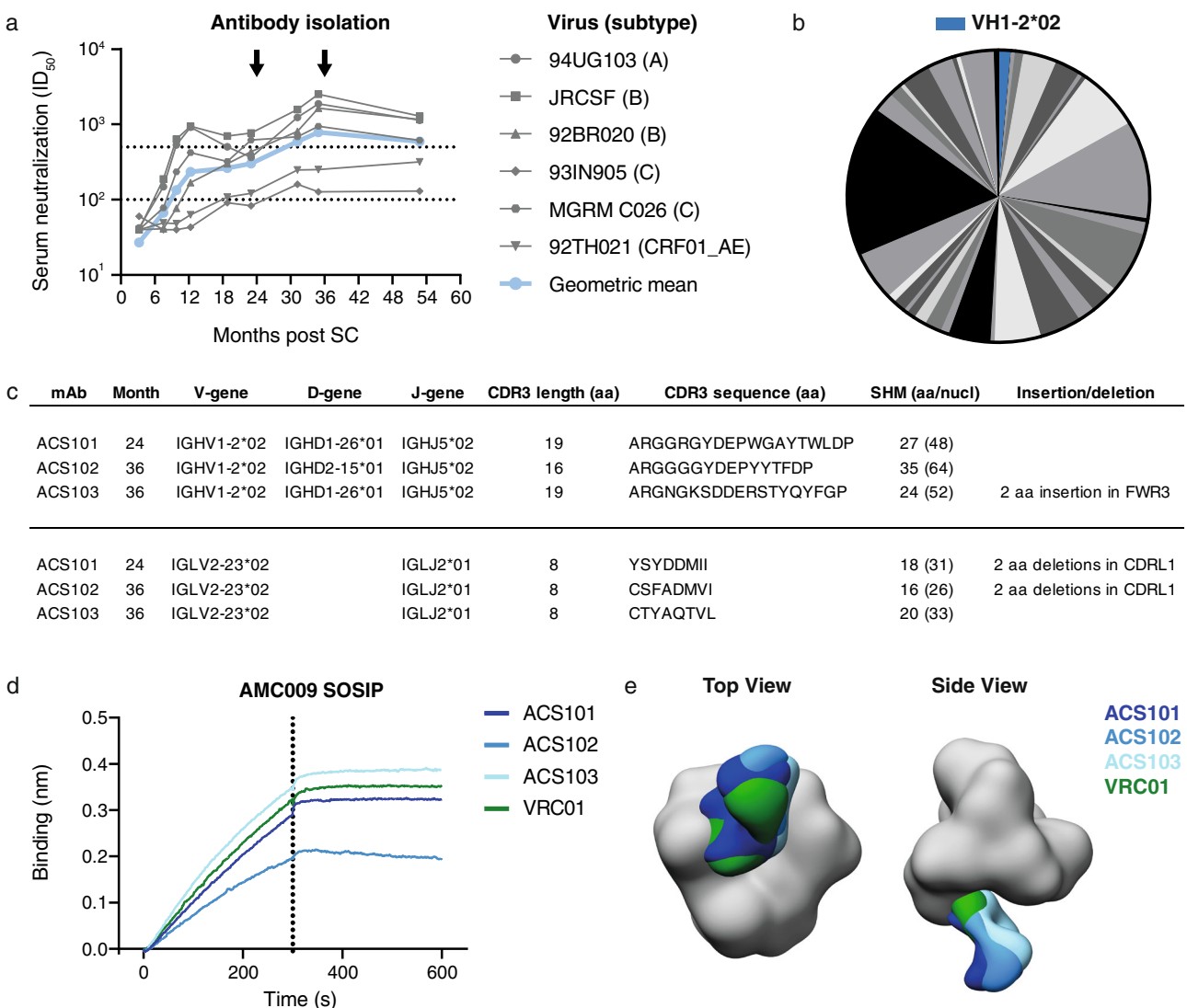

**Fig. 1 | Isolation and characterization of VH1-2-derived mAbs with average-length CDRL3 that target the CD4bs. a** Longitudinal serum neutralization (serum dilution that inhibits 50% of viral infectivity ($ID_{50}$)) of donor H18877 against a multiclade virus panel. The geometric mean is represented as a blue line. Serum samples were obtained at 3.2, 7.5, 9.8, 12.3, 18.8, 25.0, 31.2, 34.9, 52.8 months post SC, while PBMCs were isolated at month 24.0 and 36.0 post SC as indicated by the black arrows. **b** Pie-chart of VH gene usage of isolated B cells from months 24 and 36 post SC ($n = 222$). Each gray-color coded slice represents a certain VH gene and the slice size is proportional with the amount of BCRs derived from this particular VH gene. BCRs using the VH1-2*02 gene are colored in blue. **c** Sequence

characteristics of ACS101, ACS102 and ACS103 that use the VH1-2*02 and VL2-23*02 gene segments in their HC and LC, respectively. Somatic hypermutation (SHM) is depicted as the number of amino acid (aa) and nucleotide (nucl) mutations compared to the VH1-2*02 and VL2-23*02 germline genes. **d** Binding of ACS101, ACS102, ACS103, and VRC01 to AMC009 SOSIP using biolayer interferometry. **e** Negative stain electron microscopy 3D reconstructions of ACS101, ACS102, and ACS103 in complex with AMC009 SOSIP. Fabs were segmented, colored and are shown relative to a reference trimer for visualization and comparison. VRC01 (EMD-8269) Fab was included for comparison purposes.

of VRC01 as the latter is derived from a different VL gene segment (Fig. 1c).

ACS101-103 were expressed as monoclonal antibodies (mAbs) and all three mAbs bound strongly to the autologous AMC009 SOSIP trimer in biolayer interferometry (BLI) experiments, as did VRC01 (Fig. 1d). Moreover, low-resolution reconstructions from single-particle negative stain electron microscopy (ns-EM) analysis of the Fabs in complex with the AMC009 SOSIP trimer revealed that all bound the CD4bs with a similar angle of approach as VRC01 (Fig. 1e). ACS101 and ACS103 bound with a stoichiometry of three Fabs per trimer, whereas for ACS102, we only observed one or two Fabs bound to the trimer possibly the result of the low affinity of ACS102 for the AMC009 SOSIP trimer. Taken together, we identified three mAbs directed to the CD4bs that belong to the IOMA-class, and have shared sequence characteristics with other VH1-2-derived bNAbs.

## ACS101-103 possibly originated from different B cell precursors

We performed a genetic analysis to understand whether ACS101, ACS102 and ACS103 originated from a common B cell precursor or from multiple precursors. All three mAbs used the same V and J gene segments in their HC and LC. However, based on our inference, the ACS101 and ACS103 HCs used a different D gene segment compared to ACS102, suggesting that ACS102 might be clonally unrelated (Fig. 1c). In contrast, sequence comparison demonstrated that the genetic distance between ACS101 and ACS102 was shorter compared to ACS103 indicating that the ACS101 was more closely related to ACS102 compared to ACS103 (Fig. S3a, b). We then analyzed the number of shared mutations in the V, D and J regions and found that ACS101 and ACS102 shared the most mutations (Fig. S3c). Moreover, ACS101 and ACS102 share a two-residue deletion in their CDRL1. ACS101 and ACS103 did not share many mutations, while

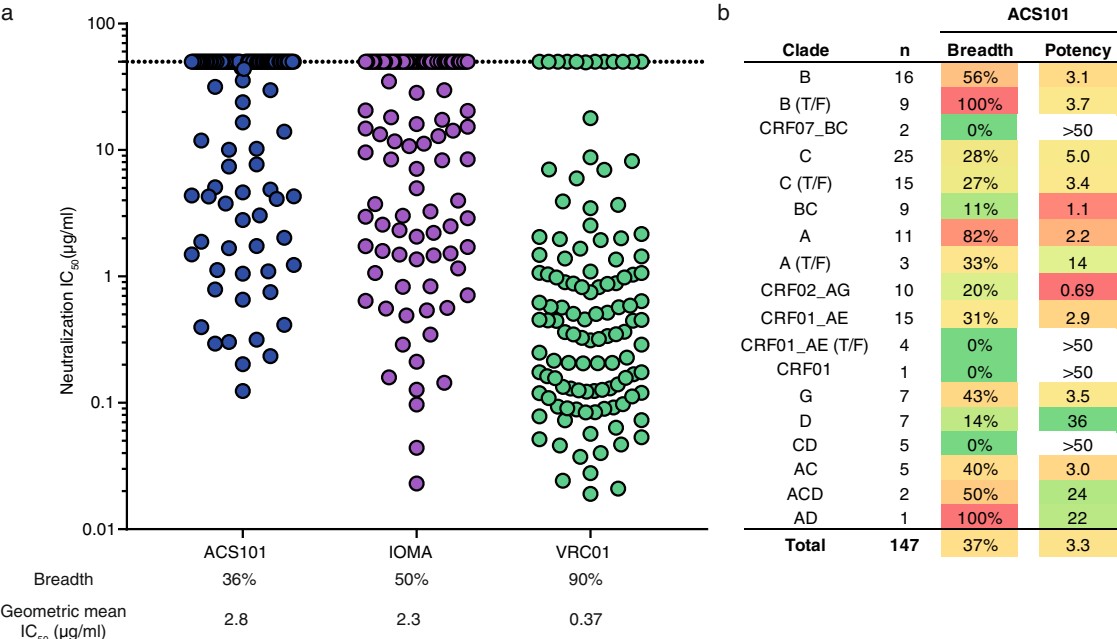

**Fig. 2 | ACS101 neutralization breadth and potency is inferior to those of IOMA and VRC01.** Breadth is indicated as the percentage of viruses that were neutralized. **a** Neutralization $IC_{50}$ profile of ACS101, IOMA and VRC01 against a panel of ($n = 119$) Tier-2 viruses. Each symbol represents a unique HIV-1 isolate. Data for IOMA and VRC01 were derived from CATNAP[78]. **b** Overview of neutralization by ACS101 against viruses from all tested panels (147 viruses in total). Breadth of ACS101 against viruses from different clades is shown with the geometric mean (potency) in µg/ml.

ACS102 and ACS103 shared the least amount of mutations. Moreover, CDRH3 and CDRL3 were substantially different between all three mAbs, which suggests that these antibodies most likely originated from different B cell precursors. However, based on these data we cannot conclusively determine whether these differences arose by SHM or whether they were different from the start (Fig. S2). Thus, all three mAbs shared mutations to some extent but also possessed distinct features, complicating the interpretation of the data and preventing assignment of true B-cell precursors. Therefore, we were unable to determine whether ACS101–ACS103 represent three unrelated IOMA-class mAbs, or whether they represent descendants of different branches originating from the same precursor. If the latter is the case, the branches must have split very early during the evolution of these mAbs.

## ACS101 exhibits moderate neutralization breadth and potency

We then tested the neutralizing activity of the mAbs ACS101 and ACS103 against the autologous AMC009 virus as well as nine viruses from the global panel, representing HIV-1's global diversity (Fig. S4a)[32]. ACS102 did not express efficiently and was not included in these assays. ACS101 and ACS103 neutralized the autologous AMC009 virus with half-maximal inhibitory concentration ($IC_{50}$) values of 0.28 and 0.054 µg/ml, respectively, and exhibited cross-neutralizing activity against several isolates of the nine-virus global panel, of which ACS101 neutralized four and ACS103 two (Fig. S4a). During affinity maturation ACS103 acquired a two amino acid insertion in its HC FWR3 (Fig. S2). ACS103 neutralizes AMC009 virus more potently but has lower neutralizing breadth compared to ACS101. This suggests that the two amino acid insertion in the HC FWR3 influences neutralization of certain genotypes more than others.

We further determined the neutralization breadth of ACS101 using additional sets of virus panels. These included the 20-virus lasso panel of which the 9-virus panel represents a subset[33], a selection of viruses from the f61 panel[34], and a 7-virus panel consisting of several non-B and non-C tier 2 viruses. ACS101 neutralized 16/43 viruses (37%) with a geometric mean $IC_{50}$ of 5.1 µg/ml, particularly neutralizing

viruses from clades A, B and C with sporadic neutralization of viruses from other clades (Fig. S5). Finally, to compare ACS101's potency and breadth with CD4bs bNAbs VRC01 and IOMA, we assayed ACS101 against a large multiclade panel of 119 viruses. ACS101 neutralized ~36% of the tested viruses with a geometric mean $IC_{50}$ of 2.8 µg/ml (Fig. 2a, Fig. S6). When combining the data from all virus panels tested (removing any virus doublets that were also part of the first panels), ACS101 neutralized 55/147 of the tested viruses (~37%) with a geometric mean $IC_{50}$ of 3.3 µg/ml (Fig. 2b). ACS101 exhibited greater breadth against clade A, clade B and clade B transmitter/founder viruses (82%, 56 and 100% breadth, respectively) than viruses from other clades. The latter finding is consistent with donor H18877 being infected with a clade B virus. The breadth and potency of ACS101 was only slightly lower to that of IOMA but substantially inferior compared to that of VRC01, consistent with IOMAs limited ability to achieve similar high levels of breadth and potency as VRC01-class bNAbs (Fig. 2a).

## Structure of ACS101 in complex with the AMC009 SOSIP trimer

To better understand the molecular details of the epitope-paratope interaction, we determined the structure of AMC009 SOSIP in complex with the ACS101 and ACS124 Fabs using single-particle cryo-electron microscopy (cryo-EM). ACS124 is a mAb that was isolated from the same HIV-1 infected individual, H18877, and was included to overcome orientation bias on the cryo-EM grid. ACS124 binds the gp41-gp120 interface (Fig. S7a), thereby leaving the CD4bs exposed for ACS101 binding. ACS124 did not neutralize any viruses from the global panel and was not studied further phenotypically (Fig. S7b). The cryo-EM complex was reconstructed to a ~4.0 Å resolution, and we also determined crystal structures of the unliganded ACS101 and ACS124 Fabs at 2.2 Å and 1.7 Å resolution, respectively (Fig. S8a–d). The complex contained three ACS101 Fabs and two ACS124 Fabs interacting with the AMC009 SOSIP trimer (Fig. S8e). CDRL1, CDRL2 and CDRH3 of the liganded ACS101 Fab in complex with AMC009 SOSIP showed subtle conformational differences compared to the unliganded ACS101 Fab, possibly due to the lower resolution of the cryo-EM

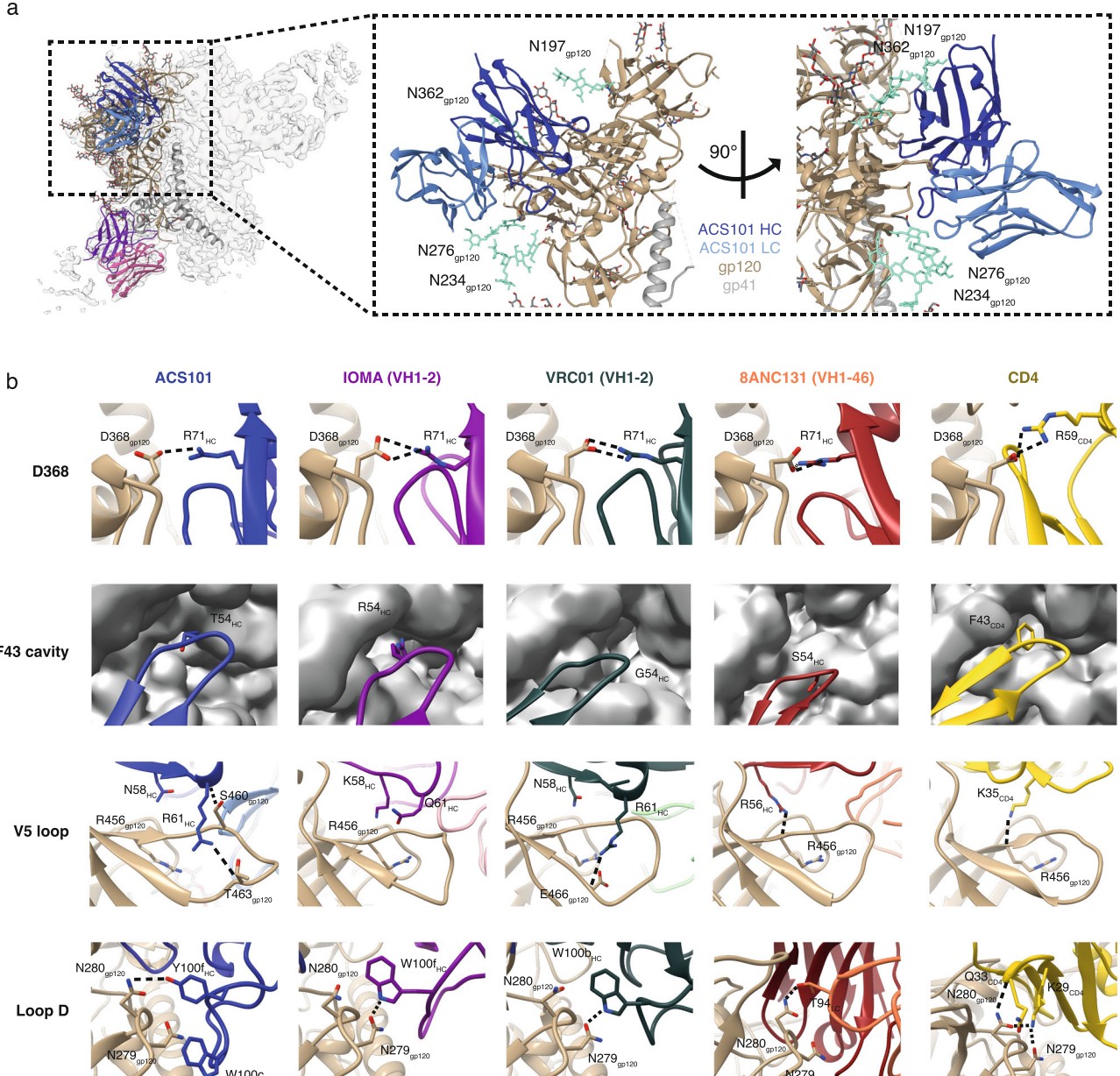

**Fig. 3 | ACS101 heavy chain interactions with the CD4bs of AMC009 SOSIP and comparison to other VH1-2 and VH1-46-derived CD4bs bNAbs. a** Cryo-EM map and model of Fabs ACS101 and ACS124 in complex with AMC009 SOSIP (gp120; light brown, gp41; gray and N-linked glycans; green). ACS101 and ACS124 Fab HC and LC are color coded as indicated. Only the variable domains of Fabs ACS101 and ACS124 and one gp41-gp120 heterodimer are shown as ribbons and fitted into the cryo-EM map. ACS101 is framed by the N197, N234, N276 and N362 glycans, which are colored green in the right-hand panel. **b** Key interactions of ACS101, VH1-2-derived bNAbs IOMA (PDB:5T3Z) and VRC01 (PDB:3NGB), VH1-46-derived bNAb

8ANC131 (PDB:4RWY) and CD4 (PDB:1G9N) with HIV-1 gp120 (shown in light brown) at four sites: D368, F43 cavity, V5 loop and Loop D. All interactions are shown with the gp120 subunit that the bNAbs were complexed with: ACS101 (clade B AMC009 SOSIP), IOMA (clade A BG505 SOSIP), VRC01 (clade A/E 93TH057 gp120), 8ANC131 (clade B YU2 gp120) and CD4 (clade B YU2 gp120). The F43 cavity is shown as a molecular map generated at 3.5 Å resolution from the gp120 atomic coordinates with UCSF Chimera[60]. Expected H-bonds and salt bridges are shown as dotted lines using a cut-off distance of 3.2 Å.

complex (~4.0 Å) compared to unliganded ACS101 crystal structure (Fig. S8c). ACS101 binds an epitope surrounding the CD4bs with a buried surface area (BSA) of 1578 Å², only 30% (484 Å²) of which is contributed by the ACS101 LC.

ACS101 is framed by the glycans at N197, N234, N276 and N362 (Fig. 3a). We did not observe any visible molecular interactions between ACS101 and the N197, N234 and N362 glycans, as only the core sugars of these glycans were resolved in the cryo-EM map. However, ACS101 clearly interacts with the N276 glycan, burying 184 Å² of surface, of which 44 Å² (23%) is by CDRH3 and 140 Å² (77%) by CDRL1. Moreover, we observed density in the cryo-EM map between ACS101

CDRL1 and CDRH3 with the N276 glycan indicating close proximity and molecular contacts (Fig. S9a). To gauge the effect of the N276 glycan on ACS101 neutralization, we tested ACS101 against N276Q mutants of the global panel viruses. The N276Q substitution rendered five of the eight viruses more sensitive to neutralization by ACS101 (TRO.11, X1632, Ce1176, Ce0217 and CNE55) (Fig. S4b). Moreover, 45_01dG5, an early virus naturally lacking the N276 glycan from the patient from which VRC01 was isolated, was highly sensitive to ACS101 neutralization[35] (Fig. S4b). These data suggest that ACS101 accommodates and makes extensive contacts with the N276 glycan, but that removal of the N276 glycans renders the core ACS101 epitope more

accessible, an observation that is also true for VRC01[9,36] (Fig. S4b) and IOMA[18].

The interactions of ACS101 with AMC009 SOSIP share similarities with those observed with other VH1-2 and VH1-46-class bNAbs (Fig. 3b). For example, ACS101 CDRH2 interacts with the CD4-binding loop on gp120, where residue $R71_{HC}$ forms a salt bridge with D368 (Fig. 3b, Fig. S9b, c). Due to the relatively low resolution of the structure, While all VH1-2 and VH1-46 bNAbs use CDRH2 for engaging the F43 cavity on gp120, the atomic details differ substantially. ACS101 contains a threonine at residue $54_{HC}$ and engages the F43 cavity similar to bNAbs VRC01 (glycine) or 8ANC131 (serine) (Fig. 3b, Fig. S2). This is in contrast to other CD4bs bNAbs, some of which penetrate the F43 cavity with hydrophobic residues at position $54_{HC}$, including VRC-PG20 ($W54_{HC}$) and 12A21 ($F54_{HC}$), or via an arginine at $54_{HC}$, including IOMA and 1B2530[8,18] (Fig. S2). Furthermore, a germline-encoded $R61_{HC}$ in VRC01 and ACS101, but not IOMA, penetrates the cavity formed by the V5 and β24-strand of gp120 (Fig. 3b, Fig. S2). ACS101 binds with a similar angle of approach as IOMA while VRC01 approaches the trimer with a slightly different angle (Fig. S9d).

## The ACS101 heavy chain exhibits a unique binding mode to N280 in loop D

VRC01-class bNAbs contact residues N279 and N280 via a tryptophan at position $100b_{HC}$ in their CDRH3 ($W100b_{HC}$), and these interactions are critical for binding and neutralization (Figs. 3b and 4a)[37,38]. On the other hand, IOMA uses a tryptophan at $100f_{HC}$ ($W100f_{HC}$) to generate similar interactions with N280 (Figs. 3b and 4a). $W100b_{HC}$ in VH1-2 bNAbs is usually located four residues (VRC01-class), or five residues in the case of $W100f_{HC}$ (IOMA-class), before the start of FWR4 (Fig. S2). ACS101 contains a tryptophan at position $100h_{HC}$ ($W100h_{HC}$) located three residues before the CDRH3 C-terminus but this residue does not contact Env. Instead, ACS101 uses a tryptophan at position $100c_{HC}$ ($W100c_{HC}$) that is located nine residues before the start of FWR4 to contact N279 (Figs. S2 and S9a). However, $W100c_{HC}$ in ACS101 is too distant from N280 and based on the density in the cryo-EM map only contacts N279 and not N280. Interestingly, $W100c_{HC}$ in ACS101, in contrast to $W100b_{HC}$ in VRC01 and $W100f_{HC}$ in IOMA, also interacts with the first *N*-acetylglucosamine of the N276 glycan (Fig. S9a). ACS101 and IOMA contain a germline-encoded $W50_{HC}$ in their CDRH2 but, unlike VRC01-class bNAbs, it does not contact N280 in loop D[18]. Furthermore, $D93_{LC}$ in CDRL3 of IOMA interacts with R456 in V5 as well as N280 in loop D, while the same residue in ACS101 only interacts with R456, not N280 (Fig. 4a). Thus, ACS101 shares contacts with IOMA, but not of VRC01-class bNAbs, to N280 in loop D.

Neutralization experiments showed that substitutions at positions 279 as well as 280 abrogated neutralization by ACS101 (N279A in Ce0217; N280D in 25710 and X1632), similar to their effects on VRC01 neutralization[35] (Fig. S4b), suggesting that the interaction of ACS101 with N280 is mediated by alternative contacts. Indeed, the structure reveals that a tyrosine at $100f_{HC}$ ($Y100f_{HC}$) overlaps with VRC01 $W100b_{HC}$ and IOMA $W100f_{HC}$ (Fig. S9e). We did not observe any density in the cryo-EM map between $Y100f_{HC}$ and N280, but $Y100f_{HC}$ could form a hydrogen bond with N280 as the distance between the sidechains is <3.2 Å (Fig. 3b, Fig. 4a). To test the role of $Y100f_{HC}$ we substituted this residue for tryptophan (Y100fW) and tested binding to AMC009 SOSIP and neutralization of the AMC009 virus. Interestingly, the ACS101 Y100fW mutant bound less efficiently to AMC009 SOSIP compared to ACS101 (Fig. S10a). In addition, ACS101 Y100fW neutralized the AMC009 virus less well, with an $IC_{50}$ that was ~12-fold higher compared to that of ACS101 (Fig. S10b). These results show that ACS101 shares HC contacts with VRC01-class bNAbs and/or IOMA, but binds to residues N279 and N280 in a different manner, using other residues at the tip of its CDRH3, in particular $W100c_{HC}$ and $Y100f_{HC}$.

## The ACS101 and IOMA light chains accommodate the N276 glycan in similar ways

ACS101 uses the VL2-23*02 gene for its LC and includes an eight-residue CDRL3, similar to IOMA. Superimposition of the structure of the AMC009 SOSIP trimer in complex with ACS101 and the same trimer in complex with the VRC01-class bNAb PGV04 with a five-residue CDRL3 illustrates a ~2.7 Å shift of the V5 loop away from the tip of the CDRL3 when AMC009 SOSIP was bound to ACS101 compared to PGV04 (Fig. 4b). This is consistent with observations made with IOMA, where the longer CDRL3 also causes a shift in the V5 loop[18]. The sequence of AMC009 SOSIP contains a potential N-linked glycosylation site (PNGS) at position 463 in the V5 loop, however we did not observe any sugars of this glycan resolved in the cryo-EM map, possibly due to the relatively low resolution of the EM map.

Affinity maturation of VRC01-class bNAbs leads to deletion of residues from CDRL1 or substitution of bulky residues for glycine, to reduce clashes with the N276 glycan[9,18] (Fig. S2). In the CDRL1 of ACS101, two residues are deleted to accommodate the N276 glycan (Fig. 4c, Fig. S2) and ACS101 CDRL1 makes contact with the N276 glycan via $S29_{LC}$ (Fig. 4c, Fig. S9a). In addition, $Y30_{LC}$ makes a backbone contact with $D92_{LC}$ in its CDRL1-CDRL3 interface (Fig. 4c). VH1-2 bNAb IOMA uses the same VL2-23*02 LC as ACS101 but its CDRL1 only acquired one additional glycine substitution and no deletions[18] (Fig. S2). As a result, IOMA's longer CDRL1 makes a short α-helix instead of an extended loop as seen in other CD4bs bNAbs (Fig. 4d). While shorter in length, the ACS101 LC also forms this short α-helix in its CDRL1 (Fig. 4d). This is particularly visible in the crystal structure of the unliganded ACS101 Fab but less so in the more loop-like CDRL1 of ACS101 Fab in complex with AMC009 SOSIP, possibly as a consequence of the lower resolution of the cryo-EM complex compared to the crystal structure of the unliganded ACS101 Fab (Fig. 4d, Fig. S8d).

The induction of substitutions or deletions in CDRL1 to accommodate the N276 glycan is one of the major roadblocks for vaccine design to elicit bNAbs to the CD4bs. To better understand the contribution of the two-residue deletion in CDRL1 of ACS101 binding and neutralization, we restored the serine and aspartic acid in CDRL1 to match the corresponding residues in the VL2-23*02 germline sequence ($ACS101_{iGL\ CDRL1}$). We also produced an $ACS101_{IOMA\ CDRL1}$ mutant that included the same CDRL1 as IOMA, including the extra glycine compared to the VL2-23*02 germline sequence. The $ACS101_{iGL\ CDRL1}$ and $ACS101_{IOMA\ CDRL1}$ mutants bound less efficiently to AMC009 SOSIP compared to ACS101 (Fig. 4e), and were also somewhat less efficient at neutralizing the AMC009 virus, although the differences were very subtle (Fig. 4f). To assess whether a five-residue CDRL3 would improve binding and neutralization of ACS101, we produced a chimeric mAb consisting of the HC of ACS101 HC and the LC of VRC01 ($ACS101_{HC}$/$VRC01_{LC}$). This chimeric mAb bound less well to AMC009 SOSIP and neutralized AMC009 with substantially reduced potency compared to ACS101 (Fig. S10b), suggesting that a five-residue CDRL3 that typifies VRC01-class bNAbs does not benefit ACS101. Overall, these data indicate that the ACS101 LC accommodates the N276 glycan and engages the V5 loop in similar ways as the IOMA LC.

## The transmitter-founder virus in donor H18877 lacked the N276 glycan

To understand the co-evolutionary process between the AMC009 virus and the ACS101 lineage, we inspected the sequences from infectious biological AMC009 virus clones sampled at various time points during infection: month 2 post SC ($n = 5$); month 8 ($n = 5$); month 10 ($n = 7$); month 13 ($n = 5$); month 18 ($n = 9$); month 27 ($n = 5$); month 36 ($n = 6$); month 49 ($n = 5$) (Fig. 5)[20]. We noted that one of the early *env* sequences isolated from donor H18877 at 2 months post SC (clone 2 M 1F9) lacked the PNGS at position 276. The presence of the N276 glycan poses a formidable challenge to the induction of CD4bs-directed

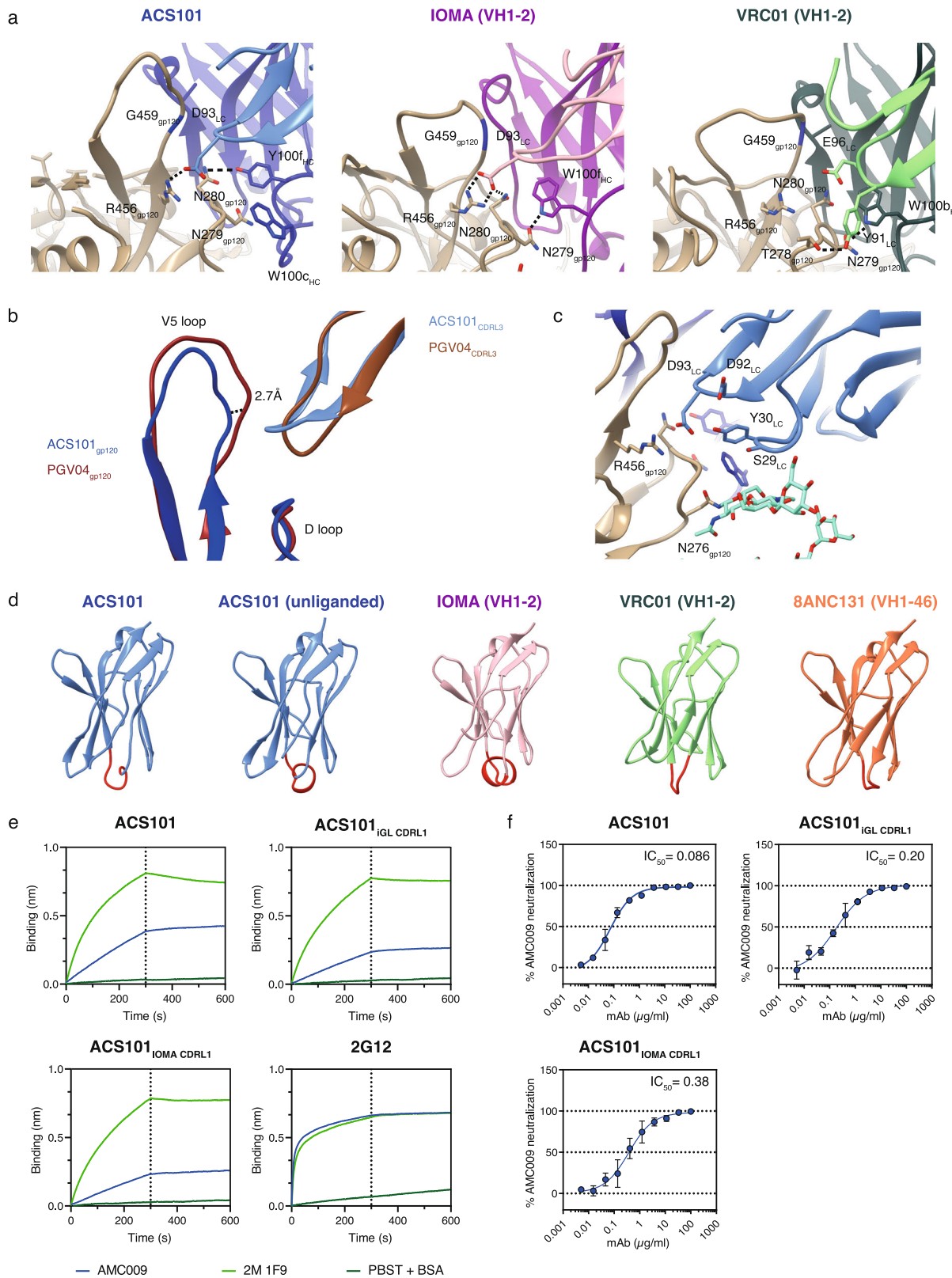

bNAbs and previous studies suggest that the VRC01 lineage was initiated by an early virus variant that lacked the N276 glycan[35]. By analogy, it is conceivable that early precursors of mAbs ACS101, ACS102, and ACS103 might have been activated by an AMC009 virus lacking the N276 glycan. However, without the actual transmitter/founder virus(es) and germline antibody (as opposed to the inferred germline antibody) we are unable to confirm this hypothesis. In all *env*

sequences isolated from later time points the M278T substitution restored the PNGS at position 276, with the exception of two *env* sequences from 36 months post SC that contained a lysine or methionine at position 278 (Fig. 5). The introduction of the N276 glycan might represent an early viral escape pathway of a virus under immune pressure from antibodies targeting the CD4bs, including early lineage members of ACS101, ACS102, and/or ACS103.

**Fig. 4 | ACS101 light chain interactions with the CD4bs of AMC009 SOSIP and comparison to other VH1-2 and VH1-46-derived CD4bs bNAbs. a** Interaction of ACS101, IOMA (PDB:5T3Z) and VRC01 (3NGB) with the V5 loop. All interactions are shown with the gp120 subunit that the bNAbs were complexed with: ACS101 (clade B AMC009 SOSIP), IOMA (clade A BG505.664 SOSIP) and VRC01 (clade A/E 93TH057 gp120). **b** Comparison of the V5 loop of ACS101 and PGV04 (PBD:6VO3) bound to gp120. Both ACS101 and PGV04 are in complex with AMC009 SOSIP. The shift of the V5 loop is indicated as a dotted line measured from Cα- Cα of residue S460. **c** Interactions of ACS101 with the V5 loop and the N276 glycan. **d** Ribbon representations Fabs (only the light chain is shown) of ACS101 and other VH1-2 and VH1-46-derived CD4bs bNAbs (liganded) with the CDRL1 colored in red. ACS101 is shown as liganded and unliganded Fab. IOMA (PDB:5T3Z), VRC01 (PDB:3NGB) and 8ANC131 (PDB:4RWY) Fabs are liganded. **e** Binding of ACS101, ACS101$_{iGL\ CDRL1}$ and ACS101$_{IOMA\ CDRL1}$ to AMC009 SOSIP and AMC009 2 M 1F9 SOSIP using biolayer interferometry (BLI). The bNAb 2G12 and PBST + BSA (PBS pH 7.4 + 0.01% (w/v) BSA + 0.002% (v/v) Tween 20) were included as a loading and negative control, respectively. **f** Neutralization of AMC009 by ACS101, ACS101$_{iGL\ CDRL1}$ and ACS101$_{IOMA\ CDRL1}$. The length of the error bars is the standard deviation at each concentration (*n* = 2 biologically independent experiments).

When assessing loop D, we observed that, in *env* sequences isolated 8 months post SC and later, N279D, N279K or N279E mutations were introduced, which likely represent escape mutations as ACS101 makes several important contacts with N279 according to our structural analyses; similar escape mutations were found in the *env* sequences isolated from the donor from whom VRC01 was isolated[35]. ACS101 neutralized viruses that included N279 or D279 (Fig. S7) suggesting that the N279D was not a resistance mutation by itself. Moreover, sequence analysis demonstrated that N279D arose concomitantly with deletions in the β23-V5 loop, suggesting that these deletions and not N279D might conferred resistance to early lineage members of ACS101 (Fig. 5). In addition, N279K has been shown to confer complete resistance to various VRC01-class antibodies[35], likely due to the change of the negatively charged aspartic acid to the positively charged side chain of the lysine that interacts with W100h$_{HC}$ (VRC01-class), or W100h$_{HC}$ in the case of ACS101.

No mutations were found at position 280, but mutations started to emerge at position 281 from month 10 onwards, and at positions 282 and 283 from month 18 onwards. (Fig. 5). The majority of sequences from month 36 and 49 had three mutations in loop D, in addition to the M278T substitution that restored the N276 glycan. Residues at positions 281–283 are not directly contacted by ACS101 and substitutions at these positions likely did not facilitate viral escape directly. However, previous studies have shown that mutations at position 281 can restore the replicative capacity of the virus to normal levels[35,39] suggesting that these mutations evolved simultaneously to compensate for escape mutations that exerted a fitness cost to viral replication.

Escape mutations at positions 364, 365, 371, and 372 were found sporadically in the CD4-binding loop sequences from months 18, 27, 36, and 49 months post SC (Fig. 5). Again, residues at these positions do not interact with ACS101 directly. Moreover, substitutions in the CD4-binding loop have shown to benefit viral fitness[40], suggesting that these substitutions were also introduced to increase viral fitness. In addition, deletions and substitutions of aa residues in the β23-V5 loop region were commonly found in *env* sequences from 2 months post SC and later time points. Shortening of the V5 loop likely influences the interaction of ACS101's CDRL1 with R456 in the V5 loop. Interestingly, sequences from 10 months post SC and later time points included an additional PNGS at position 466 that will directly clash with ACS101 LC, suggesting an additional escape mechanism of the virus (Fig. 5).

To better understand the viral escape from ACS101, ACS102, and ACS103 in donor H18877, we produced nine AMC009 SOSIPs based on longitudinal sequences (two from months 2 and 10, and one from months 8, 13, 18, 27 and 36 each), in addition to the original AMC009 SOSIP, which was based on a consensus of the month 2 sequences. We used biolayer interferometry to test binding of ACS101, ACS102, and ACS103 to these longitudinal SOSIPs. All three mAbs bound substantially better to the AMC009 Env variant that lacked the N279 glycan (2M 1F9) compared to the other AMC009 SOSIPs (Fig. 6, Fig. S10c). ACS101$_{iGL\ CDRL1}$, ACS101$_{IOMA\ CDRL1}$, ACS101$_{Y100fW}$ and ACS101$_{HC}$/VRC01$_{LC}$ also bound substantially better to the AMC009 2M 1F9 variant (Fig. 4e, S10a). ACS101, isolated from PBMCs taken at month 24 post SC, recognized AMC009 Env variants from early time points better than those

isolated from later time points during infection (Fig. 6, Fig. S10c). In addition, ACS101 did not have any affinity to Env variants from month 27 (27M 2C4) and 36 (36M 1A8) post SC indicating that by that time the virus escaped completely from ACS101. ACS102, isolated from PBMCs taken at month 36 post SC, bound better to early Env variants from month 2 (2M 1B5 and 2M 1F9) and 8 (8M 1C5) post-SC and to a lesser extent those isolated 10 months (10M 1A7 and 10M 1C11) post-SC (Fig. 6). No binding was observed to Env variants 13M 2F9, 18M 2A4 and 27M 2C4 but some affinity for Env 36M 1A8 was observed. In contrast, ACS103 isolated from PBMCs taken at month 36 post-SC bound to all Env variants including those from later time points (Fig. 6) suggesting that the virus did not escape completely from ACS103 yet. In conclusion, these data suggest that a virus lacking the N276 initiated the development of ACS101, ACS102 and ACS103 early during infection and that mutations in the binding site led to escape over time.

## Discussion

The isolation of VRC01-class bNAbs from multiple HIV-1 infected individuals has fueled the design of immunogens to elicit such responses by vaccination. These broad and potent bNAbs share common features, including the use of the VH1-2*02 germline gene segment, a short and/or flexible CDRL1, and a short five-residue CDRL3. The field has yet to elicit VRC01-class bNAbs by immunization but immunogens specifically designed to engage B cell precursors are capable of priming VRC01-class precursors in transgenic mice[13–16] and are able to engage these precursors in humans[41,42]. However, the maturation pathway toward mature VRC01-class bNAbs include extensive SHM and it remains unclear how to steer these rare B-cell precursors toward neutralization breadth after priming with a germline-targeting immunogen. The discovery and isolation of IOMA, a VH1-2 CD4-mimetic bNAb with an average-length CDRL3 and less SHM compared to VRC01, suggested an additional pathway toward the elicitation of VH1-2 CD4-mimetic bNAbs. Here, we described the isolation and characterization of multiple other IOMA-class NAbs derived from the VH1-2*02 and VL2-23*02 germline segments and including a normal-length CDRL3. Discovery of additional IOMA-class antibodies suggests they may be more common than previously appreciated. In addition, we provide data that imply that a N276-lacking transmitter/founder virus initiated and drove maturation of this lineage in the donor from whom ACS101 was isolated. The breadth and potency of ACS101 did not recapitulate serum neutralization of donor H18877, suggesting that other unidentified family members might be broader or other bNAb specificities contributed to the exceptionally broad and potent serum response of the donor.

Analysis of the human naive B cell repertoire demonstrated that eOD-GT8, an HIV-1 immunogen designed to prime VRC01-class precursors, was able to bind BCRs from diverse VRC01-class naive B cells as defined by carrying a VH1-2-derived BCR with a five-residue CDRL3[41]. In addition, IOMA-class naive B cells were also present among the eOD-GT8+ B cells from naive human donors at a frequency of 1 in 3 × 10$^6$ B cells[41]. These data suggest it should be possible to target IOMA-class precursors in the human antibody repertoire by engineered germline-targeting immunogens. However, this frequency is most likely a low estimate of the number of IOMA-like precursors found in naïve human

```
                              Loop D              CD4 binding loop           β23-V5 loop
                             (275-283)              (364-374)                 (455-472)

                         275      280          365      370          455    460            470
                          •        •            •        •            •      •              •
      H18877.2m.1F9       QNFMNNAKV            SSGGDPEIVMH           TRDGGSNENKTSE~TETFRPAG
      H18877.2m.1B5       ...T.....            ...........          ......S.S....~........
      H18877.2m.1C3       ...T.....            ...........          ............~.........
      H18877.2m.1G9       ...T.....            ...........          .......S..~~~~........
      H18877.2m.2B1       ...T.....            ...........          .......S....~.........
      H18877.8m.1C3       ...TD....            ...........          ...........E.~~~~......
      H18877.8m.1C5       ...T.....            ...........          ..........T~.~~~~......
      H18877.8m.1G4       ...T.....            ...........          ......D.SE.N.T........
      H18877.8m.2E3       ...T.....            ...........          .....~..S....~........
      H18877.8m.2F10      ...T.....            ...........          .....D.SE.N.T........
      H18877.10m.1A7      ...TD....            ...........          .........T~~~~~......
      H18877.10m.1A9      ...TD....            ...........          A.......T~~~~~......
      H18877.10m.1B3      ...T.....            ...........          .....~..SNSE~~........
      H18877.10m.1C11     ...TD....            ...........          ......D.SE.N.T........
      H18877.10m.1D10     ...T..V..            ...........          A.......T~~~~~......
      H18877.10m.1E8      ...TD....            ...........          .........E~~~~........
      H18877.10m.111      ...TD....            ...........          A.....~~.E.~~~......
      H18877.13m.2F9      ...TD.T..            ...........          .....~~~.E...~........
      H18877.13m.1D10     ...TD.T..            ...........          A.......E~~~~........
      H18877.13m.1G7      ...T..V..            ...........          A......S~~..~........
      H18877.13m.2A5      ...TD....            ...........          .....~~~.ES..~........
      H18877.13m.2E5      ...TD.T..            ...........          .....~~~.E.N.~........
      H18877.18m.1B8      ...TD....            .......E...          .....~~~...N.T........
      H18877.18m.1C10     ...TD.T..            ...........          A....~~~...N.T........
      H18877.18m.2A4      ...TD.T..            ...........          .....~~~..NN.T........
      H18877.18m.2A6      ...TD....            ...........          A.......SD.~~~~......
      H18877.18m.2C5      ...T..VR.            ...........          ......D.SEA~~~~......
      H18877.18m.2C7      ...TD....            ...........          A...~~~...N.T........
      H18877.18m.2C12     ...TD.V..            ...........          .....~~~...N.T........
      H18877.18m.2F9      ...TD.T..            ...........          .....~~~...N.T........
      H18877.18m.2G11     ...TD.T.I            ...........          .....~~~...N.T........
      H18877.27m.1A7      ...TK.D..            .......E..           .....~~~...N.T.....E.
      H18877.27m.1C11     ...TK.D..            ...........          .....~~~...N.T.....E.
      H18877.27m.1F6      ...TK.D..            ...........          .....~~~...N.T.....E.
      H18877.27m.2C4      ...TK.D..            .......E..           .....~~~...N.T~.......
      H18877.27m.1G1      ...TK.D..            ...........          A....~~~...N.T........
      H18877.36m.1A8      ...KK.DR.            .......E..           .....~~~...N.~........
      H18877.36m.1B10     ...TK.DR.            L..........          ...NKT.E.~.T........
      H18877.36m.1F10     ...TK.T..            .P.........          .....~~~...N.T........
      H18877.36m.1G9      ...TK.DR.            .......E..           .....~~~...N.T~.......
      H18877.36m.2E1      ....E.DG.            ...........          ...NKT.E.~.T........
      H18877.36m.2F6      ...TK.DG.            ...........          .....~~~...N.T.....E.
      H18877.49m.1A12     ...TE.DG.            .T.........          .....~~~.T.N.T~....I.
      H18877.49m.1C1      ...T..N.I            ...........          .....~~~.T.N.T~....V.
      H18877.49m.1C3      ...TK.T..            .T.........          .....~~~.T.N.T~....V.
      H18877.49m.1D6      ...TE.DG.            .T.........          .....~~~.T.N.T~....I.
      H18877.49m.1G2      ...TD.N.I            ...........          .....~~~.T.NKT~.......
```

**Fig. 5 | Sequence alignment of longitudinal Envs from donor H18877 for three sites, e.g., Loop D, CD4-binding loop, and the β23-V5 loop.** Residues that are contacted by ACS101 revealed by the cryo-EM structure are highlighted in blue. The *env* sequence are named as followed; donor_name.month.clone, e.g. H18877.2 m.1F9 indicates the sequence 1F9 from donor H18877 at month 2 post SC. Symbol (-) indicate a deletion at that particular position.

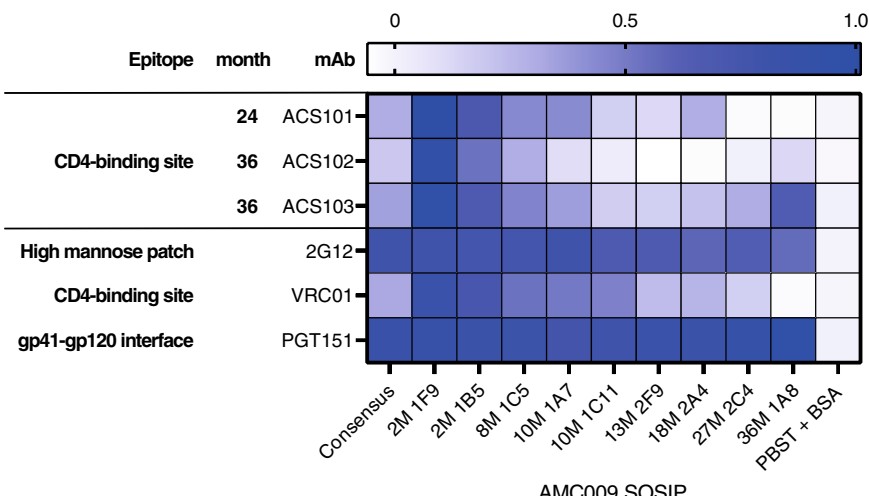

**Fig. 6 | Binding of ACS101, ACS102, and ACS103 to longitudinal Envs isolated from donor H18877.** Binding of ACS101, ACS102, and ACS103 to longitudinal Envs isolated from donor H18877 using biolayer interferometry. Heat map representing the maximum values of the binding curves measured by biolayer interferometry. The bNAbs 2G12, VRC01 and PGT151 were included as controls, whereas PBST + BSA (PBS pH 7.4 + 0.01% (w/v) BSA + 0.002% (v/v) Tween 20) was used as a negative control.

individuals because eOD-GT8 does not bind the inferred germline of IOMA with detectable affinity[41]. The development of immunogens specifically designed for optimal engagement of IOMA-class precursors bNAbs would therefore likely further increases the chances of inducing IOMA-class bNAbs. Such germline-targeting immunogens may include modifications to select for IOMA-class precursors with a short α-helix in CDRL1, normal-length CDRL3 and specific residues at the tip of the CDRH3, all of which contribute to the NAb activity of IOMA and ACS101. The isolation and characterization of ACS101, ACS102 and ACS103 and the structure of ACS101 in complex with the AMC009 trimer might facilitate such efforts.

We show that one of the early viruses isolated from donor H18877 harbored an Env lacking the N276 glycan and that ACS101 bound with higher affinity to this Env, suggesting that the lack of this glycan initiated the development of the ACS101 lineage. Almost all *env* sequences isolated from later time points possessed a PNGS sequon at position 276. These observations suggest that prime-boost immunization strategies to elicit IOMA-class bNAbs might include priming immunogens lacking N276 and boosting immunogens containing the N276 glycan to increase neutralization breadth. An immunization strategy based on the longitudinal *env* sequences from donor H18877 might represent one sequential vaccination approach to induce IOMA-class antibodies.

In summary, the discovery of additional IOMA-class NAbs combined with identification of the longitudinal *env* sequences that drove their initiation and maturation provides important knowledge for the design of vaccines aimed at inducing IOMA-class bNAbs.

## Methods
### Patient samples
PBMCs were obtained from a male participant, donor H18877, who was enrolled in the MSM (men having sex with men) component of the Amsterdam Cohort Studies on HIV/AIDS[19,20]. The Amsterdam Cohort Studies are conducted in accordance with the ethical principles set out in the Declaration of Helsinki, and written consent was obtained prior to data collection. The study was approved by the Academic Medical Center Institutional Medical Ethics Committee. Individual H18877 was infected with a subtype B HIV-1 variant and seroconverted during active follow-up. He did not receive combination antiretroviral therapy during the period when PBMCs and/or sera were collected. During this period, he had a detectable viral load and his CD4 T-cell count was stable. In addition, H18877 was homozygous for the CCR5 gene and had no protective HLA type.

The longitudinal development of the HIV-1-specific neutralizing activity in individual H18877 has been previously reported, and H18877 was assigned there as individual C1[20]. The AMC009 SOSIP trimer was based on a consensus sequence derived from the *env* sequences of five clonal isolates obtained at two months post-SC[21]. The *env* sequences used to make the longitudinal AMC009 SOSIP trimers were derived from clonal isolates obtained at various months post-SC[20], and were produced as SOSIPv4.2 trimers. The AMC009 virus was generated from one of the five viral clones isolated at two months post-SC and differs by three amino acids compared to the AMC009 SOSIP trimer in addition to the stabilizing SOSIP mutations and a potential N-linked glycosylation site (PNGS) at position 339[21].

### Fluorescence-activated cell sorting, single B cell (RT)-PCR and cloning
B cells were sorted from PBMCs taken at months 24 and 36 post-SC, as described follows[24]. PBMCs were stained in (phosphate-buffered saline) PBS supplemented with 1 mM EDTA and 1% FBS at 4 °C for 60 min as follows. The cells were stained with fluorophore-conjugated anti-human CD3, CD8, CD14, CD19, CD20, CD27, IgG and IgM (BD Pharmigen). In addition, B cells from month 24 post-SC were stained with biotinylated AMC009 SOSIP, BG505 SOSIP and 94UG103 gp120 Env proteins conjugated to Streptavidin-PE (Life Technologies), Streptavidin-APC (Life Technologies) and Streptavidin-BV785 (BioLegend), respectively. Memory B cells (CD19$^+$/CD20$^+$/IgG$^+$/IgM$^-$/HIV$^+$) that were AMC009 SOSIP$^+$/94UG103 gp120$^+$ or could bind all three Env proteins were single cell sorted. In contrast, B cells from month 36 post-SC were stained with biotinylated MGRM C026 gp120, 94UG103 gp120 and BG505 SOSIP coupled to Streptavidin-PE (Life Technologies), Streptavidin-BV785 (BioLegend) and Streptavidin-APC (Life Technologies), respectively. Memory B cells (CD19$^+$/CD20$^+$/IgG$^+$/IgM$^-$/HIV$^+$) that could recognize at least one of the three Env proteins were single cell sorted. For both sorting strategies, unwanted cell populations (CD3$^-$/CD8$^-$/CD14$^-$) were excluded. HIV-1 Env-specific B cells were single cell sorted into 96-well plates containing lysis buffer using a BD FACSAria III machine. Plates were stored at −80 °C.

mRNA was reverse-transcribed and polymerase chain reaction (PCR) amplified to generate HC and LC variable V(D)J regions to produce mAbs as follows[22,30,43]. HotStar Taq DNA polymerase master mix (Qiagen) was used in all PCR reactions in combination with primer sets that have been described elsewhere[43]. HC and LC regions were sequenced and analyzed with the IMGT V-quest webserver[44,45].

The selected VH1-2*02-derived HCs and the corresponding VL2-23*02-derived LCs were cloned into the corresponding Igγ1 and Igλ expression vectors, respectively, as described previously[30].

## Phylogenetic sequence analysis

The HC and LC sequences were analyzed with IMGT HighV-QUEST to identify the framework, CDR regions and VDJ gene segments[46]. The amino acid distances between the framework and CDR regions of all antibodies were calculated by all versus all pairwise global alignments using the "alignment" Python package[47]. Multiple sequence alignments were performed for each chain with the germline and ACS101-103 sequences using Clustal Omega[48]. A custom python script was used to count mutations compared to germline and to determine the number of shared mutations between the ACS101-103 sequences. The percentage of shared mutations was calculated as the number of shared mutations divided by the total number of mutations in each sequence.

## Monoclonal antibody production

All ACS101 variants were made using the Q5 Site-Directed Mutagenesis Kit (New England Biolabs). MAbs were produced by co-transfection of plasmids encoding for the HC and LC (1:1 ratio) into HEK293F cells (Invitrogen, cat no. R79009) using 1 mg/ml PEImax (Polysciences Europe GmBH, Eppelheim, Germany). HEK293F cells (Invitrogen, cat no. R79009) were maintained at a density of ~1 million cells/ml in FreeStyle Medium (Life Technologies) prior to transfection. The supernatant was harvested five days after transfection, centrifuged and filtered using 0.22 μm pore size Steritops (Millipore, Amsterdam, The Netherlands). The supernatant was then flowed (0.5–1.0 ml/min) over a protein G agarose (Pierce) affinity chromatography column, washed with PBS and eluted with 0.1 M glycine pH 2.5 into 1 M Tris pH 8. MAbs were buffer exchanged into PBS using Vivaspin20 centrifugal filters with 100 kDa MW cut-off (Sartorius, Göttingen, Germany).

## Monoclonal antibody digestion

Monoclonal IgGs were digested to Fab by incubating the IgGs with freshly-activated papain for 5 h at 37 °C. Papain was activated in digestion buffer (100 mM Tris pH 8, 2 mM EDTA, 10mM L-cysteine) for 15 min at 37 °C prior to antibody digestion. 50 mM iodoacetamide was added to stop the reaction and the digestion products were buffer exchanged into tris-buffered saline TBS. Fabs were purified by size exclusion chromatography using TBS and fractions corresponding to the Fab peak were pooled and concentrated using a Amicon concentrator when necessary.

## AMC009 Env production

AMC009 SOSIPv5.2 and SOSIPv4.2 trimers were produced as follow[49]. In brief, HEK293F cells (Invitrogen, cat no. R79009) were maintained at a density of ~1 million cells/ml in FreeStyle Medium (Life Technologies) before transfection. 293F cells were co-transfected with two plasmids encoding the AMC009 SOSIP trimer or furin using 1 mg/ml PEImax (Polysciences Europe GmBH, Eppelheim, Germany). The supernatant was harvested six days after transfection, centrifuged and filtered using 0.22 μm pore size Steritops (Millipore, Amsterdam, The Netherlands). The supernatant was then flowed (0.5–1.0 ml/min) over a PGT145 affinity column at 4 °C. The column was then first washed with PBS followed by a second wash with 20 mM Tris-HCl pH 8.0, 0.5 M NaCl. SOSIPs trimer were eluted from the column with 3 M MgCl₂ pH 7.2, 20 mM Tris-HCl into an equal volume of 20 mM Tris-HCl pH 8.0, 75 mM NaCl. All SOSIP trimers were buffer exchanged into TBS, concentrated, and further purified by size exclusion chromatography in TBS.

## Neutralization assay

Neutralization assays with the global panel viruses, global panel N276/N280/N279 mutants, small panel consisting of several non-B and non-C tier 2 viruses, virus panel f61 and the multiclade panel of 119 viruses were performed as follows. NAbs were measured as a function of reduction in luciferase (Luc) reporter gene expression after a single round of infection in TZM-bl cells[50,51]. TZM-bl cells (also called JC57BL-13) were obtained from the NIH AIDS Research and Reference Reagent Program, as contributed by John Kappes and Xiaoyun Wu. This HeLa cell clone was engineered to express CD4 and CCR5[52] and to contain integrated reporter genes for firefly luciferase and *E. coli* beta-galactosidase under control of an HIV-1 LTR[53]. Briefly, a pre-titrated dose of virus was incubated with serial 5-fold dilutions of mAbs (50 μg/ml starting concentration) in a total volume of 150 μl for 1 h at 37 °C in 96-well flat-bottom culture plates. Freshly trypsinized cells (10,000 cells in 100 μl of growth medium containing DEAE dextran) were added to each well. One set of control wells received cells + virus (virus control) and another set received cells only (background control). After 48 h of incubation, 100 μl of cells was transferred to a 96-well black solid plate (Costar) for measurements of luminescence using the Britelite Luminescence Reporter Gene Assay System (PerkinElmer Life Sciences). Neutralization titers are the concentration at which relative luminescence units (RLU) were reduced by 50% compared to virus control wells after subtraction of background RLUs. Assay stocks of molecularly cloned Env-pseudotyped viruses were prepared by transfection in 293 T/17 cells (American Type Culture Collection) and titrated in TZM-bl cells as described[50,51]. Global panel N276/N280/N279 mutations were introduced into env clones by site-directed mutagenesis. All mutations were confirmed by full-length env gene sequencing. This assay has been formally optimized and validated[54] and was performed in compliance with Good Clinical Laboratory Practices, including participation in a formal proficiency testing program[55]. Additional information on the assay and all supporting protocols may be found at: http://www.hiv.lanl.gov/content/nab-reference-strains/html/home.htm.

Neutralization assays with ACS101, ACS101$_{iGL\ CDRL1}$, ACS101$_{IOMA\ CDRL1}$, ACS101$_{Y100fW}$, ACS101$_{HC}$/VRC01$_{LC}$ and VRC01 against AMC009 were performed in-house as follows[50,56]. In brief, mAbs were measured as a function of reduction in luciferase (Luc) reporter gene expression after a single round of infection in TZM-bl cells (obtained through the NIH AIDS Reagent Program, Division of AIDS, NIAID, NIH from Dr. John C. Kappes, and Dr. Xiaoyun Wu). TZM-bl cells were seeded at ~17.000 cells (37.5 μl medium/well) in half-area 96-well plates. The mAbs were threefold serial diluted (50 μg/ml starting concentration) and were then incubated with previously titrated Env-pseudotyped viruses for 1 h at room temperature (RT). Assay stocks of molecularly cloned Env-pseudotyped viruses were prepared by transfection in 293Tcells and titrated in TZM-bl cells. In addition, we also included wells containing virus or cells only as a positive and negative control, respectively. DEAE (40 μg/ml) and sanquinvir (400 nM) were added to TZM-bl-cells before the cells were incubated with the mAb/virus mixture for three days at 37 °C. Luciferase activity was measured using the Bright-Glo (Promega) and GloMax Discover System and IC$_{50}$ were calculated using GraphPad Prism version 9.0.

## Biolayer interferometry

The biolayer interferometry experiments using longitudinal SOSIP variants was performed as follows. All experiments were performed in reaction buffer (PBS pH 7.4 + 0.01% (w/v) BSA + 0.002% (v/v) Tween 20) at room temperature (RT) using an Octet Red96 instrument (ForteBio). Anti-human IgG (AHI) probes were first equilibrated in reaction buffer for 60 s. IgGs were diluted to 10 μg/ml in reaction buffer and immobilized onto the AHI probes for 300 s, followed by a wash for 60 s in reaction buffer. The binding of SOSIPv4.2 trimers to the IgGs was then measured at a concentration of 250 nM for 300 s, followed by dissociation for 300 s in reaction buffer. Analysis was performed using Octet software and GraphPad Prism version 9.0. Biolayer interferometry experiments using the ACS101$_{Y100fW}$, ACS101$_{HC}$/VRC01$_{LC}$,

ACS101$_{iGL\ CDRL1}$ and ACS101$_{IOMA\ CDRL1}$ variants were performed similarly as described above, but with protein A (Fortebio) biosensors.

## Negative stain electron microscopy

Fabs or IgGs were complexed with SOSIP trimers for 30 min at RT with a 4-fold and 2-fold molar excess of Fab and IgG, respectively. Fab/IgG-SOSIP complexes were diluted to 30 ng/μl in TBS and loaded onto glow-discharged, carbon-coated Cu400 EM grids for 10 s followed by blotting to remove excess sample. The Fab/IgG-SOSIP complexes were then stained with 2% (w/v) uranyl formate and immediately blotted, followed by a second stain with 2% (w/v) uranyl formate for 30 s before blotting to remove excess stain. Image collection was performed on a Tecnai Spirit T12 (FEI) (120 kV, ×52,000 magnification) equipped with an Eagle 4 K CCD (FEI/Thermo Fisher) camera or Tecnai T20 (FEI) (200 kV, ×62,000 magnification) equipped with a TemCam F416 CMOS (TVIPS) camera using Leginon. Data processing was performed as follows[57,58]. 2D classification and 3D sorting was performed using Relion v3.0[59] and EM maps were visualized and segmented using UCSF Chimera[60] and Segger[61], respectively.

## Cryo-electron microscopy data collection

Three hundred micrograms of AMC009 SOSIPv5.2 trimer was incubated with sixfold molar excess of ACS101 and ACS124 Fab for 1 h at RT. The complex was SEC purified in TBS to remove excess Fab and fractions corresponding to the AMC009 SOSIP-ACS101-ACS124 complex were pooled and concentrated using an Amicon concentrator (Millipore) to 6.0 mg/ml for application onto cryo-EM grids. To prepare the cryo-EM grids we used a Vitrobot mark IV (Thermo Fisher Scientific) with the chamber set to 10 °C and the humidity was maintained at 100%. The blotting time was varied between 6.5 and 7 s, wait time was set to 10 s and no blot force was applied to the cryo-EM grids. The AMC009 SOSIP-ACS101-ACS124 complex was mixed with either lauryl maltose neopentyl glycol (LMNG) at a final concentration of 0.005 mM or n-Dodecyl-β-D-Maltopyranoside (DDM) at a final concentration of 0.06 mM before application to Quantifoil (R 1.2/1.3, 400) grids. Quantifoil (R 1.2/1.3, 400) grids were plasma cleaned with Ar/O$_2$ (Solarus plasma cleaner, Gatan) for 10 s before sample application. After sample application and blotting, the grids were plunge-frozen into liquid ethane. The grids were loaded into a FEI Talos Arctica electron microscope (ThermoFisher) operating at 200 keV and data was collected using Leginon software[57]. The microscope was equipped with a K2 Summit direct electron director camera (Gatan). Magnification was set to 36,000 with a resulting pixel size of 1.15 Å. Micrograph movie frames were aligned and dose-weighted using MotionCor2[62]. GCTF was used to determine CTF models[63]. Particle picking, 2D classification, Ab-initio reconstruction, and 3D refinement were conducted using cryoSPARC v3.2.0[64]. Additional information regarding data collection and processing of the AMC009-ACS101-ACS124 complex can be found in Table S1.

Initial Fab models were generated using SAbPred[65] and docked together with the Env-portion of PDB: 6VO3 into the EM map using UCSF Chimera[60]. Additional mutations for SOSIPv5.2 compared to SOSIPv4.2 (PDB: 6V03) were introduced using Coot[66,67], and N-linked glycans were added[68]. Iterative rounds of manual model building in Coot[66,67] and RosettaRelax[69,70] were performed to refine the models into the EM map. The final models were analyzed and evaluated with MolProbity[71] and EMringer[72]. jsPISA was used to calculate buried surface areas (average of all three binding sites) for ACS101 Fab[73]. Distance measurements were performed using PDBePISA[74] with a cut-off for expected H-bonds and salt bridges of 3.2 Å.

## Protein production, and crystallization

For crystallography, ACS101 and ACS124 Fabs were produced in transiently transfected Expi293F cells and then purified using CH1-XL affinity matrix beads (Thermo Scientific), followed by size exclusion chromatography using a Superdex75 16/60 column (GE Healthcare) with running buffer TBS; 20 mM Tris, 150 mM NaCl, pH 7.4. Fabs ACS101 and ACS124 entered crystallization trials using our automated CrystalMation robotic system (Rigaku) at Scripps Research at concentrations of 10.4 and 9.5 mg/ml, respectively. Crystals of ACS101 Fab were obtained at 20 °C with a precipitant solution of 10% (v/v) glycerol, 0.1 M of HEPES (pH 7.5), 5% (w/v) polyethylene glycol 3000 and 30% (v/v) polyethylene glycol 400. Crystals of ACS124 Fab were obtained at 20 °C with a precipitant solution of 0.1 M of CHES (pH 9.5) and 38% (v/v) polyethylene glycol 600. Crystals were cryo-protected with reservoir crystallization solution followed by fast plunging into liquid nitrogen and storing before data collection.

## X-ray data collection, data processing and structure determination

Diffraction data for ACS101 and ACS124 Fabs were collected at Advanced Light Source (at Lawrence Berkeley National Laboratory) beamline 5.0.1 and at Stanford Synchrotron Radiation Lightsource (SSRL) beamline 12-1, respectively. All data processing was performed using HKL2000[75]. Molecular replacement for ACS124 was performed using coordinates from germline precursor of 3BNC60 Fab[37] (PDB ID: 5F7E, 1.90Å) where the CDR loops were removed as search models in Phaser[76]. Molecular replacement for ACS101 was performed using coordinates from IOMA-class CLK31 Fab[41] (PDB ID: 6D2P, 2.60Å) where the CDR loops were removed as search models in Phaser. All crystal structures were refined using Phenix.refine[76]. The model building was done in Coot[77]. Data collection and refinement statistics for all crystal structures are reported in Table S2.

## Reporting summary

Further information on research design is available in the Nature Research Reporting Summary linked to this article.

# Data availability

Atomic coordinates and structure factors of the reported crystal structure have been deposited in the Protein Data Bank (PDB: 7U04, PDB: 7U0K). Cryo-EM reconstructions have been deposited in the Electron Microscopy Data Bank (EMD-14474) and in the Protein Data Bank (PDB: 7Z3A). The negative stain 3D EM reconstructions are deposited to the Electron Microscopy Data Bank (EMD-14475, EMD-14476, EMD-14477 and EMD-14478). The accession numbers for ACS101-103 and ACS124 BCR sequences are DDBJ/ENA/GenBank: OM744384- OM744391. The neutralization and biolayer interferometry data generated in this study are provided in the Supplementary Information. Source data are provided with this paper.

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

## Acknowledgements

This work was supported by the HIV Vaccine Research and Design (HIVRAD) program (P01 AI110657 to A.B.W., R.W.S), the Bill and Melinda Gates Foundation CAVD (OPP1132237 to R.W.S., INV002022 to R.W.S. and OPP1146996 to M.S.S.), NIH/NIAID (HHSN272201800004C to D.C.M), and the European Union's Horizon 2020 research and innovation program under grant agreement no. 681137 (R.W.S.). R.W.S. is a recipient of a Vici fellowship from the Netherlands Organization for Scientific Research (NWO). J.v.S. is a recipient of a 2017 AMC Ph.D. Scholarship. This research used resources of the SSRL, SLAC National Accelerator Laboratory, which is supported by the U.S. Department of Energy, Office of Science, Office of Basic Energy Sciences under Contract No. DE-AC02-76SF00515. The SSRL Structural Molecular Biology Program is supported by the DOE Office of Biological and Environmental Research, and by the National Institutes of Health, National Institute of General Medical Sciences (including P41GM103393). Beamline 5.0.1 of the Advanced Light Source, a U.S. DOE Office of Science User Facility under Contract No. DE-AC02-05CH11231, is supported in part by the ALS-ENABLE program funded by the National Institutes of Health, National Institute of General Medical Sciences, grant P30 GM124169-01.

## Author contributions

J.v.S., R.W.S., A.B.W., M.J.v.G. conceived the experiments; J.v.S., D.S., D.C.M., D.R.B., M.S.S., G.O., I.A.W., R.W.S., A.B.W., M.J.v.G. contributed to the design of the study and analysis of the data; J.v.S., A.S., T.L.G.M.v.d.K., T.G.R.M. designed and produced SOSIP trimers; M.J.v.G., D.S. performed B-cell sorting; J.v.S., B.D.C.v.S., A.H.C.v.K. performed B-cell receptor sequence analysis; J.v.S., T.L.G.M.v.d.K., P.v.d.W., J.A.B., T.B., R.G., M.J.v.G., performed antibody cloning and production; J.v.S., performed negative stain electron microscopy studies; J.v.S., H.L.T., G.O. performed the cryo-electron microscopy studies; E.F., R.L.S., I.A.W. solved and analyzed crystal structures; J.v.S., H.G., J.D., C.C.L., D.C.M., M.S.S. performed and analyzed in vitro neutralization assays; J.v.S., performed biolayer interferometry experiments; J.v.S., R.W.S., M.J.v.G. wrote the manuscript; All authors were consulted for reviewing relevant components of the manuscript.

## Competing interests

The authors declare no competing interests.
