## [Peer Review File · Nature Communications]

REVIEWER COMMENTS

Reviewer #1 (Remarks to the Author):

This is a very clear, well written paper describing the identification of CD4-BS targeting antibodies on HIV-1 env that resemble the previous described IOMA neutralizing antibody. They report the isolation, characterization (binding, neutralization breadth) and structure of these novel antibodies, in particular of ACS101. The authors also evaluated the role of the CDRL1 by making some chimeric antibodies and measuring binding and neutralization. They also sequenced the virus within the patient and suggest that a transmitter/founder lacking glycan at N276 initiated the development of these Mabs.

This paper should be of interest to the HIV antibody/vaccine community but the authors could review the previous literature with more details and maybe discuss this finding in a broader context. Some experiments could be done - such as affinity measurements for example - see detailed comments below.

First the authors could describe what the IOMA-class is. Is it a CD4-BS antibody with a VH1-2*02 paired with a VL2-23*02 with CDRL3 of 8 aa?

What if VRC01-class and IOMA were all in a VH1-2*02 class ? How will the immunogens designed for IOMA-class differ from the ones designed for VRC01-class, yes IOMA-like have longer CDRL3 but is this difference what really matters? The authors could discuss this further.

This was alluded to in the following paper that the authors cite:
<https://www.science.org/doi/pdf/10.1126/scitranslmed.aat0381>

The authors did not cite the Bonsignori et al, Immunity, 2018 paper which indicated that VH1-2*02 VRC01-like antibody with longer CDRH3 compensates for longer CDRL1. It seems an oversight that people forget to consider the CDRH3 of these antibodies. It has been shown that the CDRH3 matters.

This is surprising that glycans in V5 are not discussed and the proximity to the N-terminus of the LC.

2m.1B5 has glycan at N276... so unclear which virus started the lineage and the authors should mention this - how do they know for sure the lineage did not start at month 2?

Minor things:

-Would be nice to state breadth of the antibodies in abstract/intro.

-Can the authors discuss why ACS103 is more potent but less broad.

-Line 223 - this observation is also true for VRC01, what about IOMA? The authors could refer to these 2 mAbs.

“but that removal of the N276 glycans renders the core ACS101 epitope more accessible.”

-A workflow for cryoEM data processing is missing in supplemental

Is there a reason why C3 symmetry was not applied for the CryoEM processing (or both C1 and C3).

-Fig 1C - line 827

Somatic hypermutation (SHM) is determined as the percentage of nucleotide differences with the VH1-2*02 and VL2-23*02 germline genes.

Is it % or number of mutations in the figure (unclear) ?

-Fig 1e - what about IOMA?

Binding orientation - is it the same as IOMA or not?

-What about affinity measurement with Fab? It would be more informative than a yes or no binding.

-Change that statement “ACS101 neutralization breadth and potency is comparable to IOMA but inferior to VRC01” - it is also inferior to IOMA...

-Figure 3 - line 843 color coded and not colored code

Hard to see that only N276 interact - ACS101 is framed by the N197, N234, N276 and N362 glycans

-Figure 4b - why comparing to PGV04 - because Same trimer? AMC009 SOSIP-PGV04? Ok but what about other VRC01 mab?

- The V5 loop in Fig 4c could be shown better (with the arg 456)

-Fig 4e - why is PBS/BSA binding to 2G12?

Please measure affinity

-Line 868 - panel f only

-Figure 6 - what about affinity - looks like similar on rate and off rate.

-Line 287 - Rewrite this sentence - the roadblock is not the ability, is it? “The ability of VRC01-class bNAbs to accommodate the N276 glycan by substitutions or deletions in CDRL1 is one of the major roadblocks for induction of bNAbs to the CD4bs by vaccination

Fig 5 - is there something wrong with your alignment of the V5 loop? NKT.E similar to NKTSE sequence..

-How do the authors know they chose the right rotamer for D368 ...can the authors show a map? The one shown in fig s9 is pretty poor. Or discuss possibility for other conformations of the side chain.

- Hard to see but Thr 54 seems to be pointing away from the “Phe43” cavity - is this correct?
 - Line 228-229, would remove “and as such does not include a hydrophobic side chain at position 54HC, “ you do mention this in the sentence after.
 - N280 description not really seen in the main figure 4b.
 - Line 252-253 - “Neutralization experiments showed that substitutions at positions 279 as well as 280 abrogated neutralization by ACS101 (N279A in Ce0217; N280D in 25710 and X1632)”
- Maybe but there are likely many other mutations in these Env - doing the experiment with the authors mutating only these 2 will be a better way to support their claim.
- Does AMC009 SOSIP have a glycan or two in V5 (for example at 460?)

Reviewer #2 (Remarks to the Author):

Summary

The manuscript “Identification of novel IOMA-class neutralizing antibodies targeting the CD4-binding site on the HIV-1 envelope glycoprotein” by van Schooten et al structurally and biochemically characterizes 3 newly isolated IOMA-class bNAbs from an HIV-1 infected patient. The structures represent the first characterization of additional members within the IOMA-class of bNAbs, which represent a crucial target for HIV vaccine design. Therefore, I believe this work is of critical importance to the field and warrants publication in this journal. The authors only need to address the following minor issues listed below.

Minor Issues:

- 1) Although ACS101-3 share V and J gene segments (and ACS101 and ACS103 share D gene segments), the CDRH3s are very different, which suggests that these antibodies most likely originated from different B cell precursors. This is also implied by the neutralization data – ACS101 (isolated at 24 months post-SC) is more broad/potent than ACS102-3 (isolated at 36 months post-SC). It seems unlikely that ACS102-3 would acquire mutations to make it less broad and potent compared to ACS101 (even

though ACS103 was 5-fold better against the AMC009 Env, which is somewhat representative of the circulating virus at the time ACS101-3 were isolated from the donor described in this study)

2) Conformational changes in HIV antibodies between the bound/unbound states are pretty rare and I'm actually not aware of any published examples of structural rearrangements within CD4bs bNAbs upon binding Env_{gp120}. Generally the apo (unbound) Fab structures are preconfigured and do not make any conformational changes upon binding Env_{gp120} and I believe this would be the first example of such a case. I wonder if the ACS101 CDRL1 is modeled incorrectly due to the lower resolution of the EM maps within this region? And in reality looks more like the CDRL1 in the unliganded ACS101 Fab crystal structure (Figure 4d)? I'm assuming the authors modeled in the ACS101 Fab crystal structure into the EM maps as an initial starting model to build their structure – how well does the crystal structure fit into the density maps within CDRL1?

3) Page 17, line 391 – “IOMA-class naive B cells were also present among the eOD-GT8+ B cells from naive human donors, and at a fairly high frequency of 1 in 3×10^6 B cells.”

I'm not sure 1/3,000,000 should be considered high frequency. In addition, this number is probably a low estimate of the number of IOMA-like precursors found in a given naive human individual since the immunogen used in this experiment, eOD-GT8, did not bind to the inferred germline of IOMA with detectable affinity. The number of IOMA-like precursors identified would almost certainly be higher if using an immunogen that had higher affinity to the inferred germline of IOMA.

We appreciate the time and effort the reviewers spent to review our manuscript. We are thankful for the constructive feedback, which will improve the quality of the manuscript. We tried to address each of the reviewers comments. The reviewers text is shown in italic black, our replies are in blue.

Reviewer #1 (Remarks to the Author):

This is a very clear, well written paper describing the identification of CD4-BS targeting antibodies on HIV-1 env that resemble the previous described IOMA neutralizing antibody. They report the isolation, characterization (binding, neutralization breadth) and structure of these novel antibodies, in particular of ACS101. The authors also evaluated the role of the CDRL1 by making some chimeric antibodies and measuring binding and neutralization. They also sequenced the virus within the patient and suggest that a transmitter/founder lacking glycan at N276 initiated the development of these Mabs.

This paper should be of interest to the HIV antibody/vaccine community but the authors could review the previous literature with more details and maybe discuss this finding in a broader context. Some experiments could be done - such as affinity measurements for example - see detailed comments below.

We thank the reviewer for the accurate summary, kind words, and emphasizing the implications of this study to the field.

*First the authors could describe what the IOMA-class is. Is it a CD4-BS antibody with a VH1-2*02 paired with a VL2-23*02 with CDRL3 of 8 aa?*

In the introduction, we now clearly describe that IOMA is derived from the VH1-2*02 and VL2-23*02 germlines and possesses an average-length eight residue CDRL3 (lines 83-84). Since IOMA was the only member of this class so far, the exact properties of the IOMA-class bNAbs were hard to determine since these were based on only one member, IOMA itself. We now added "of this class" in (line 93) to illustrate that the identification of additional IOMA-like antibodies is necessary to determine the exact properties of IOMA-class bNAbs.

*What if VRC01-class and IOMA were all in a VH1-2*02 class? How will the immunogens designed for IOMA-class differ from the ones designed for VRC01-class, yes IOMA-like have longer CDRL3 but is this difference what really matters? The authors could discuss this further.*

This was alluded to in the following paper that the authors cite: <https://www.science.org/doi/pdf/10.1126/scitranslmed.aat0381>

This is an interesting discussion point that we have now incorporated in the discussion (lines 410-416).

*The authors did not cite the Bonsignori et al, Immunity, 2018 paper which indicated that VH1-2*02 VRC01-like antibody with longer CDRH3 compensates for longer CDRL1. It seems an oversight that people forget to consider the CDRH3 of these antibodies. It has been shown that the CDRH3 matters. This is a valid argument and we have now cited the Bonsignori et al. article in the introduction (lines 75-77).*

This is surprising that glycans in V5 are not discussed and the proximity to the N-terminus of the LC. AMC009 SOSIP has a PNGS at position 463 (HxB2 numbering) but we did not observe any density of a glycan in the map of the AMC009 SOSIP-ACS101 complex, most likely due to the relatively low resolution of the EM map. We have now added this to the text (lines 285-288).

2m.1B5 has glycan at N276... so unclear which virus started the lineage and the authors should mention this - how do they know for sure the lineage did not start at month 2?

We agree with the reviewer that we cannot conclude for sure that a N276-lacking virus initiated the development of these antibodies. We regret that this was not clear from the text. We have tried to write as delicately as possible that the N276-lacking 2m.1F9 virus 'might' have initiated the development of these lineages (lines 327-328, 419-421). To clarify this we have now also added in the text that we cannot rule out that a transmitter/founder virus with a PNGS at position N276 initiated the development of these antibodies (lines 327-328).

Minor things:

-Would be nice to state breadth of the antibodies in abstract/intro.

We have now added the breadth of ACS101 to the abstract and introduction (line 46 and 97).

-Can the authors discuss why ACS103 is more potent but less broad.

We now discuss why ACS103 might be more potent but less broad compared to ACS101 (lines 184-188).

-Line 223 - this observation is also true for VRC01, what about IOMA? The authors could refer to these 2 mAbs.

"but that removal of the N276 glycans renders the core ACS101 epitope more accessible."

We now refer to VRC01 and IOMA here (lines 233-234)

-A workflow for cryoEM data processing is missing in supplemental

We have now added the different steps of data processing we conducted using cryoSPARC in the methods section of the manuscript (lines 595-596).

Is there a reason why C3 symmetry was not applied for the CryoEM processing (or both C1 and C3).

We choose to apply C3 symmetry for the cryoEM processing since ACS124 Fab bound with a stoichiometry of two Fabs per trimer.

-Fig 1C - line 827

*Somatic hypermutation (SHM) is determined as the percentage of nucleotide differences with the VH1-2*02 and VL2-23*02 germline genes.*

Is it % or number of mutations in the figure (unclear)?

We now clarify that we determined SHM as the number of amino acid/nucleotide mutations compared to the VH1-2*02 and VL2-23*02 germline genes (lines 855-856).

-Fig 1e - what about IOMA?

Negative-stain data is published of VRC01 but not of IOMA. At the time that we performed the negative-stain experiments we did not have IOMA IgG or Fab in our laboratory, so we were unable to perform negative-stain experiments with IOMA. Since we make such elaborate comparisons between the high-resolution structures of ACS101 and IOMA, we find it unnecessary to perform negative-stain experiments with IOMA as we believe that this will not add any more information to the manuscript. We have now added an extra subfigure Fig. S9d where we compare surface representations of the crystal structures of IOMA and VRC01 with ACS101 (line 245-246).

Binding orientation - is it the same as IOMA or not?

See comment above. We have now added an extra subfigure Fig. S9d where we compare the binding orientation using surface representations of the crystal structures of IOMA and VRC01 with ACS101 (line 245-246).

-What about affinity measurement with Fab? It would be more informative than a yes or no binding.

We agree that affinity measurements of the antibodies would be more informative and to pin-point the binding mechanism of these mAbs even better. However since the AMC009 SOSIP and

longitudinal SOSIP trimers did not contain any tags we were restricted to first couple the IgG's to the bio-layer interferometry sensors instead of the SOSIP trimers. Therefore, we were unable to determine affinities using bio-layer interferometry set-up because of potential avidity affects. To assess the affinities of these IgGs it would require a significant amount of additional work by cloning all of the SOSIP trimers into new vectors that contain tags and create additional Fab constructs. However, we think that affinity data is not essential to the message in this manuscript that the virus in this individual evolved over time to escape antibody pressure and therefore the significant additional work is not justified.

-Change that statement "ACS101 neutralization breadth and potency is comparable to IOMA but inferior to VRC01" - it is also inferior to IOMA...

We agree with the reviewer and now state that ACS101 neutralization breadth is inferior to IOMA and VRC01 (lines 201-203, 862).

-Figure 3 - line 843 color coded and not colored code

We changed "colored code" to "color coded" in the text (line 872).

Hard to see that only N276 interact - ACS101 is framed by the N197, N234, N276 and N362 glycans

We agree that the interactions are difficult to see in Fig. 3A, however because only the core sugars of these glycans were resolved in the cryoEM map, with which no interactions were seen, we can only speculate about interactions and therefore decided not to make any changes in the figures. In the text we explain that we did not observe any visible molecular interactions between ACS101 and N198, N234, N276 and N361 (line 221-223)

-Figure 4b - why comparing to PGV04 - because Same trimer? AMC009 SOSIP-PGV04? Ok but what about other VRC01 mab?

Yes, we only made the comparison to PGV04 because this was the only structure available of a VRC01-class bNAb in complex with the AMC009 SOSIP trimer. We agree with the reviewer that it would be nice to compare the CDRL3 of more VRC01-class mAbs and their position to the V5 to the ACS101-AMC009 complex. We performed the analysis with other structures available with VRC01-class bNAbs prior to submitting the manuscript, however decided not include it as these structures were with different Env subtypes and we think that the length of the loop (and specific mutations) might affect its position more than just the antibody binding, making the comparison with ACS101 less informative.

- The V5 loop in Fig 4c could be shown better (with the arg 456)

We adjusted Fig 4c to have the V5 loop (including R456) better visible.

-Fig 4e - why is PBS/BSA binding to 2G12?

We always have some background issues when we use the protA sensors and 2G12 (Fig. 4E and S10A) but not with the anti-human IgG sensor, where 2G12 shows similar background as the other antibodies tested (Fig. 6 and Fig. S10c). We do not know exactly why 2G12 has a higher background with the protA sensors, however 2G12 was only used as a loading control to check whether similar amounts of trimers are loaded we decided to perform the chimeric antibody experiments with protA sensors. This background issue only for 2G12 is not important for the message we want to convey and therefore have not re-ran the experiments using anti-human IgG sensors (Fig. 4E and Fig. S10A).

Please measure affinity

Please see the previous comment about the affinity measurements where we explain why we did not measure any affinities.

-Line 868 - panel f only

We corrected the error in the text (line 897)

-Figure 6 - what about affinity - looks like similar on rate and off rate.

Please see the previous comment about the affinity measurements where we explain why we did not measure any affinities.

-Line 287 - Rewrite this sentence - the roadblock is not the ability, is it? "The ability of VRC01-class bNAbs to accommodate the N276 glycan by substitutions or deletions in CDRL1 is one of the major roadblocks for induction of bNAbs to the CD4bs by vaccination

We corrected this sentence (lines 301-302).

Fig 5 - is there something wrong with your alignment of the V5 loop? NKT.E similar to NKTSE sequence..

We have now adapted the alignment so that NKT.E aligns well with the NKTSE sequence, this did not change the conclusions drawn from the alignment.

-How do the authors know they chose the right rotamer for D368 ...can the authors show a map? The one shown in fig s9 is pretty poor. Or discuss possibility for other conformations of the side chain.

We have added a figure (Fig. S9B) where we show the map of the D368R interaction (line 237).

-Hard to see but Thr 54 seems to be pointing away from the "Phe43" cavity - is this correct?

We agree that Thr54 seems to be pointing away from the Phe43 cavity in Fig. 3b. However, this is not the case. We changed the angle of the image and now this is better visible.

-Line 228-229, would remove "and as such does not include a hydrophobic side chain at position 54HC, " you do mention this in the sentence after.

We removed the sentence "and as such does not include a hydrophobic side chain at position 54HC, " from the text.

-N280 description not really seen in the main figure 4b.

We have adapted the N280 description in Fig. 4b so that it is better visible now.

-Line 252-253 - "Neutralization experiments showed that substitutions at positions 279 as well as 280 abrogated neutralization by ACS101 (N279A in Ce0217; N280D in 25710 and X1632)"

Maybe but there are likely many other mutations in these Env - doing the experiment with the authors mutating only these 2 will be a better way to support their claim.

We regret that this was not clear from the text, but we only mutated the N279A or N280D in these viruses and compared neutralization to the wild-type viruses. This data is shown in Fig. S4b.

-Does AMC009 SOSIP have a glycan or two in V5 (for example at 460?)

AMC009 SOSIP has a PNGS at position 463 (HxB2) numbering but we did not observe any density of a glycan in the map of the AMC009 SOSIP-ACS101 complex. This was possibly the result of the relatively low resolution of the EM map. We have now added this to the text (lines 285-288).

Reviewer #2 (Remarks to the Author):

Summary

The manuscript "Identification of novel IOMA-class neutralizing antibodies targeting the CD4-binding site on the HIV-1 envelope glycoprotein" by van Schooten et al structurally and biochemically characterizes 3 newly isolated IOMA-class bNAbs from an HIV-1 infected patient. The structures represent the first characterization of additional members within the IOMA-class of bNAbs, which represent a crucial target for HIV vaccine design. Therefore, I believe this work is of critical

importance to the field and warrants publication in this journal. The authors only need to address the following minor issues listed below.

We thank the reviewer for recognizing the implications of this study for the field and for providing constructive feedback to improve the manuscript.

Minor Issues:

1) Although ACS101-3 share V and J gene segments (and ACS101 and ACS103 share D gene segments), the CDRH3s are very different, which suggests that these antibodies most likely originated from different B cell precursors. This is also implied by the neutralization data – ACS101 (isolated at 24 months post-SC) is more broad/potent than ACS102-3 (isolated at 36 months post-SC). It seems unlikely that ACS102-3 would acquire mutations to make it less broad and potent compared to ACS101 (even though ACS103 was 5-fold better against the AMC009 Env, which is somewhat representative of the circulating virus at the time ACS101-3 were isolated from the donor described in this study)

We agree with the reviewer that ACS101, ACS102 and ACS103 likely originated from different B cell precursors. We now have stated this more clearly in the text (lines 168-171). However, since we only have the sequences of these three mAbs and not those of any other lineage members, we find that we cannot make any definite conclusions regarding their B cell precursors.

2) Conformational changes in HIV antibodies between the bound/unbound states are pretty rare and I'm actually not aware of any published examples of structural rearrangements within CD4bs bNAbs upon binding Envgp120. Generally the apo (unbound) Fab structures are preconfigured and do not make any conformational changes upon binding Envgp120 and I believe this would be the first example of such a case. I wonder if the ACS101 CDRL1 is modeled incorrectly due to the lower resolution of the EM maps within this region? And in reality looks more like the CDRL1 in the unliganded ACS101 Fab crystal structure (Figure 4d)? I'm assuming the authors modeled in the ACS101 Fab crystal structure into the EM maps as an initial starting model to build their structure – how well does the crystal structure fit into the density maps within CDRL1?

This is a valid argument and we now have removed the text about the structural rearrangements upon ligand binding within the ACS101 Fab (lines 217-219 and 399-300). Instead, we now mention that this is possibly the result of the lower resolution of the cryo-EM complex compared to the resolution of the ACS101 Fab crystal structure. We actually used SAbPred (Dunbar, J. et al. SAbPred: a structure-based antibody prediction server. *Nucleic Acids Res.* 44, W474 (2016)) to generate initial Fab models to build the structure because the Fab crystal structures were not yet available at that time. Later when the Fab crystal structures were solved we fitted the crystal structures in the EM map to check how well the crystal structures fitted. However, for some of the regions our modeled Fab structures fitted better into our density maps compared to the crystal structures. We recognize that that might be the result of the differences in resolution between the structures as the reviewers mentions.

3) Page 17, line 391 – “IOMA-class naive B cells were also present among the eOD-GT8+ B cells from naive human donors, and at a fairly high frequency of 1 in 3×10^6 B cells.” I'm not sure 1/3,000,000 should be considered high frequency. In addition, this number is probably a low estimate of the number of IOMA-like precursors found in a given naive human individual since the immunogen used in this experiment, eOD-GT8, did not bind to the inferred germline of IOMA with detectable affinity. The number of IOMA-like precursors identified would almost certainly be higher if using an immunogen that had higher affinity to the inferred germline of IOMA.

We removed “at a fairly high frequency” and now clearly describe that eOD-GT8 does not bind the inferred germline of IOMA, and that 1/3,000,000 probably is a low estimate of the real number of IOMA-like precursors in naive individuals (line 410-416).

REVIEWERS' COMMENTS:

Reviewer #1 (Remarks to the Author):

The authors have addressed the comments. Could it be that the glycan occupancy in the V5 is not 100% rather than the result of low cryoEM resolution?

Reviewer #2 (Remarks to the Author):

The authors have more than satisfied all of the reviewer's comments. Therefore, this article should be published in Nature Communications.